# VEFill: accurate and generalizable deep mutational scanning score imputation across protein domains

Polina V Polunina [ID][1], Wolfgang Maier [ID][1] & Alan F Rubin [ID][2,3,4]✉

## Abstract

Deep Mutational Scanning (DMS) assays can systematically assess the effects of amino acid substitutions on protein function, but many datasets have incomplete variant coverage due to technical constraints. We developed VEFill (Variant Effect Fill), a gradient boosting model for imputing missing DMS scores across protein domains. Trained on the Human Domainome 1, VEFill integrates ESM-1v sequence embeddings, evolutionary conservation (EVE scores), amino acid substitution matrices, and physicochemical descriptors. The model achieved robust predictive performance (Pearson r = 0.80) and generalized reliably to unseen proteins in stability-based datasets, while showing weaker performance on activity-based assays. Per-protein models confirmed VEFill's effectiveness under limited-data conditions and a reduced two-feature version performed comparably to the full model, suggesting an efficient alternative. Across multiple benchmarking settings, VEFill consistently outperformed baselines once ≥20% of experimental measurements were available. However, true zero-shot prediction without positional context remains challenging, particularly for functionally complex proteins. Overall, VEFill offers an interpretable, scalable framework for DMS score imputation, and enables systematic mutation prioritization including the design of sparse experimental libraries for variant effect studies.

**Keywords** Deep Mutational Scanning; DMS Score Imputation; Machine Learning; Protein Stability; Variant Effect Prediction
**Subject Category** Computational Biology

## Introduction

Deep mutational scanning (DMS) is a high-throughput experimental technique that systematically assays the functional consequences of thousands of protein variants in parallel, generating detailed variant effect (VE) maps (Fowler and Fields, 2014; Fowler et al, 2023). These maps are essential for elucidating protein structure-function relationships, predicting the impact of missense mutations, and interpreting genetic variation in both research and clinical contexts. As of 2025, MaveDB (Rubin et al, 2025)—a curated repository for multiplexed assays of variant effects (MAVEs)—lists over 2600 publicly available datasets, including more than 1100 for human proteins. Although the technology is advancing rapidly, most DMS datasets remain incomplete due to experimental limitations such as low coverage, sequencing depth constraints, and assay dropout, leaving many mutations without experimental scores. This missing data hinders large-scale integrative analyses and downstream applications in variant prioritization and protein engineering.

Several supervised and unsupervised models have been developed to impute missing values in DMS datasets by leveraging evolutionary, biochemical, or structural features. Random forest-based approaches (Mighell et al, 2018; Weile et al, 2017) use averaged or position-weighted features including BLOSUM62 scores (Henikoff and Henikoff, 1992) and physicochemical differences, achieving reasonable accuracy on small numbers of curated proteins. Gradient boosting models like Envision (Gray et al, 2018) extended this framework with additional conservation and structural metrics. Deep learning methods, such as Deep2Full (Sruthi and Prakash, 2020), have incorporated neural networks with sequence and co-evolutionary features, while FUSE (Yu et al, 2024) introduced a shrinkage estimation pipeline using James-Stein-derived position-level means. Other recent methods like AALasso and FactorizeDMS (Fu, 2023) have explored variant similarity and latent matrix decomposition, respectively. Other approaches focus on imputation based solely on DMS scores without incorporating external features. For example, a recent method imputes missing scores by identifying the median of the most similar amino acid substitutions based on BLOSUM100 similarity within the same protein context, followed by refinement using weighted averaging with observed values (Gebbia et al, 2024a,b). While these models vary in feature sets and complexity, most are either limited in scalability, restricted to a few datasets, or tailored to specific proteins or assay types.

Despite growing interest in DMS score imputation, existing models often have limited generalizability due to being trained on small or protein-specific datasets, or relying on features that are not uniformly available across proteins such as structural annotations (Fu, 2023; Gray et al, 2018; Sruthi and Prakash, 2020; Weile et al, 2017). Imputation approaches may also be optimized for individual proteins (Mighell et al, 2018; Weile et al, 2017) and primarily leverage substitution matrices (e.g., BLOSUM62) or conservation-

[1]Bioinformatics Group, Department of Computer Science, University of Freiburg, Freiburg, Germany. [2]Bioinformatics Division, The Walter and Eliza Hall Institute of Medical Research, Parkville, VIC, Australia. [3]Department of Medical Biology, The University of Melbourne, Melbourne, VIC, Australia. [4]Collaborative Centre for Genomic Cancer Medicine, The University of Melbourne, Melbourne, VIC, Australia. ✉E-mail: alan.rubin@unimelb.edu.au

based scores (e.g., SIFT (Ng and Henikoff, 2001), PROVEAN (Choi et al, 2012)). While effective in certain contexts, these features largely capture broad evolutionary constraints and may fail to reflect more complex dependencies, such as long-range residue interactions, that influence mutational effects. Recent work on CPT-1 (Jagota et al, 2023)—a cross-protein transfer learning model for clinical variant classification—demonstrated strong performance across genes by combining features such as EVE scores (Frazer et al, 2021), amino acid properties, and embeddings from ESM-1v—a transformer-based language model trained on protein sequences that captures long-range dependencies via its attention mechanism (Meier et al, 2021). Although CPT-1 can be adapted for other tasks, it is primarily designed to classify variants as pathogenic or benign. This highlights a remaining need for a scalable and robust framework tailored specifically to the task of DMS score imputation.

To address this gap, we developed VEFill, a scalable and generalizable model for DMS score imputation that integrates diverse biologically informed features. Trained on the Human Domainome 1 dataset (Beltran et al, 2025)—a large, standardized dataset based on a uniform stability assay—VEFill is designed to generalize across unseen proteins and variant types without relying on structural annotations (Fig. 1). In addition to the DMS data, the model incorporates EVE scores as evolutionary constraints, amino acid physicochemical properties, amino acid substitution matrices, and ESM-1v sequence embeddings, the latter of which are designed to capture long-range dependencies within the amino acid sequence and improve predictions in a context-sensitive manner.

# Results

## Overview of Human Domainome 1 dataset

To build VEFill, we leveraged DMS data from the Human Domainome 1 dataset, which provides high-throughput mutational stability measurements across 522 protein domains. All domains were evaluated using a consistent experimental platform—an abundance-based protein fragment complementation assay (aPCA)—under standardized cellular conditions. Mutational libraries were generated using site-saturation mutagenesis, introducing all possible amino acid substitutions across each domain. After excluding one artificial sequence not present in MaveDB, 521 domains were available for downstream analysis (see Appendix Fig. S1 for an overview of data organization and feature storage, and Datasets EV1 and EV2 for dataset metadata and input feature definitions, respectively).

From this set, we selected a subset of 140 domains (comprising 136,854 mutations) for training feature-complete models using all available data types, including EVE scores, substitution matrices, physicochemical amino acid properties, and sequence-based embeddings. This well-controlled and feature-complete subset enabled the development of VEFill as a domain-generalizable model for accurate DMS score imputation under stability-focused experimental conditions.

The full 521-domain set (comprising 562,208 mutations) was used to train reduced-feature models based solely on ESM-1v embeddings and mean DMS scores. By training reduced-feature models on this broader dataset, we explored VEFill's performance and scalability in more practical, feature-limited scenarios.

## General cross-protein model evaluation

To investigate the influence of different feature sets on DMS score imputation, we trained a series of general cross-protein models on 136,854 mutations from 140 protein domains in the Human Domainome 1 dataset using a leave-protein-out strategy, where variants from 90% of proteins were used for training and variants from the remaining 10% were held out for testing (Fig. 2; Appendix Fig. S2, Dataset EV3). We evaluated performance across various biologically motivated feature combinations, including evolutionary scores (EVE), substitution matrices, physicochemical properties, and ESM-1v sequence embeddings. Simpler models using individual feature types—such as only EVE scores, substitution matrices, or biochemical descriptors—exhibited limited performance ($R^2 < 0.15$). Incorporating ESM-1v embeddings led to a substantial improvement ($R^2 = 0.42$), highlighting the utility of learned contextual sequence representations. A model relying solely on mean DMS scores per position, without any additional external features, performed better than these minimal configurations and even outperformed the model using only ESM-1v embeddings, achieving an $R^2$ of 0.53. However, it still underperformed relative to the feature-set configuration that combined mean DMS with all non-ESM features ($R^2 = 0.61$). The model incorporating both mean DMS and ESM-1v embeddings ($R^2 = 0.63$) achieved better performance than all aforementioned configurations, indicating that learned embeddings and position-specific priors capture complementary information. These results further demonstrate that non-ESM features cannot fully compensate for the contextual information provided by ESM-1v in the cross-protein setting, where models must generalize to protein domains entirely absent from the training data. The positional mean DMS score is not available for unseen proteins, making sequence-derived representations essential.

The best-performing model, which included all features except the ESM-1v difference vector, achieved a very modest improvement of $R^2 = 0.64$ and a Pearson correlation of $r = 0.80$ (Fig. 3), suggesting that most of the relevant signal from sequence embeddings was captured by wild-type and variant ESM-1v representations alone (see Dataset EV2 for abbreviations used for feature sets). These findings underscore the importance of integrating biologically relevant features—particularly learned embeddings and positional priors—for accurate, cross-domain prediction of mutational effects.

Because the density of variants is highest near the neutral score range, we additionally examined how prediction error varies across the spectrum of DMS scores. We computed RMSE in quantile-based bins across the full range of both observed and predicted normalized DMS scores (Fig. EV1). When binning by observed scores, we observed a characteristic U-shaped error profile: RMSE is lowest in the neutral region, where the training data are densest, and increases toward both the deleterious and high-activity tails. This pattern is consistent with known assay-dependent score compression and increased measurement uncertainty in extreme regions, which reduce the effective dynamic range of DMS experiments.

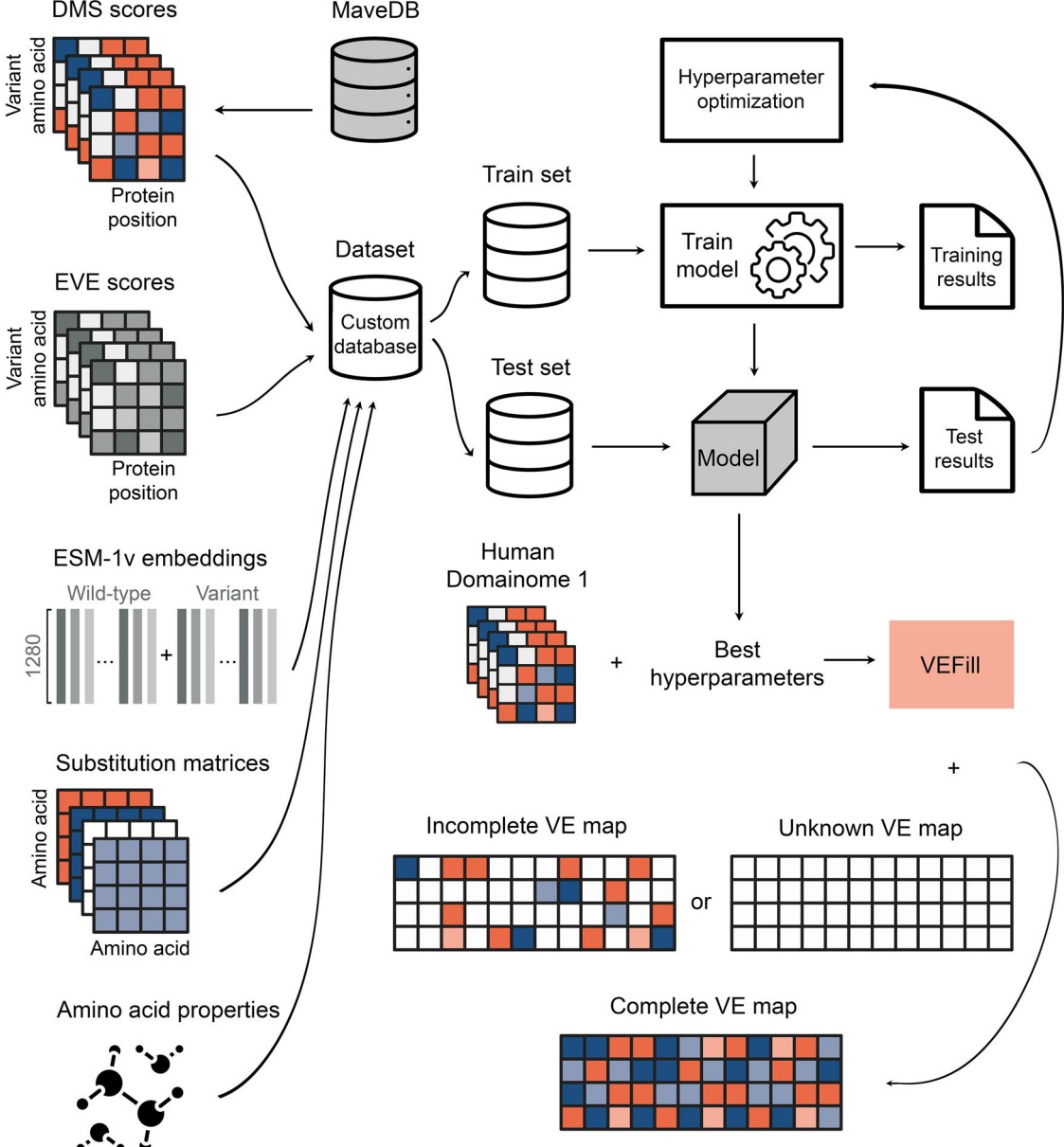

**Figure 1. VEFill model design, optimization, training, and application.**

Features collected include experimental DMS scores from human protein domains (obtained from MaveDB), EVE scores, ESM-1v embeddings, amino acid substitution matrices, and various physicochemical amino acid properties. All datasets required for model training, testing, and inference were stored in a PostgreSQL database with a custom schema. A LightGBM gradient-boosting model was trained and optimized using Bayesian hyperparameter optimization (via Optuna). The trained VEFill model can effectively impute missing DMS scores and predict variant effects, including for stability-based assays on previously unseen full-length proteins in a zero-shot manner.

We further computed RMSE as a function of predicted scores to assess model calibration (Fig. EV1). In this setting, RMSE is lowest for variants predicted to be neutral, increases for intermediate predicted scores (approximately 0.5–0.7), and decreases again toward the most deleterious predictions. Predicted high-activity variants exhibit error levels comparable to those of mildly deleterious variants. Extremely deleterious and high-activity DMS scores are predicted more conservatively and closer to the neutral region, a behavior that is common in supervised regression models trained on datasets strongly enriched for near-neutral variants. This effect is consistent with the pattern observed in Fig. 3, where predictions exhibit a characteristic compression toward the mean. Together, these results indicate that VEFill is most challenged by intermediate-effect variants, which represent a heterogeneous mixture of functional outcomes and are known to exhibit higher assay noise. In contrast, VEFill provides reliable quantitative estimates for mildly deleterious and mildly high-activity variants, resulting in RMSE values comparable to those observed in the neutral regime. Predictions at the extreme tails should therefore be interpreted as conservative approximations rather than precise estimates of extreme effects.

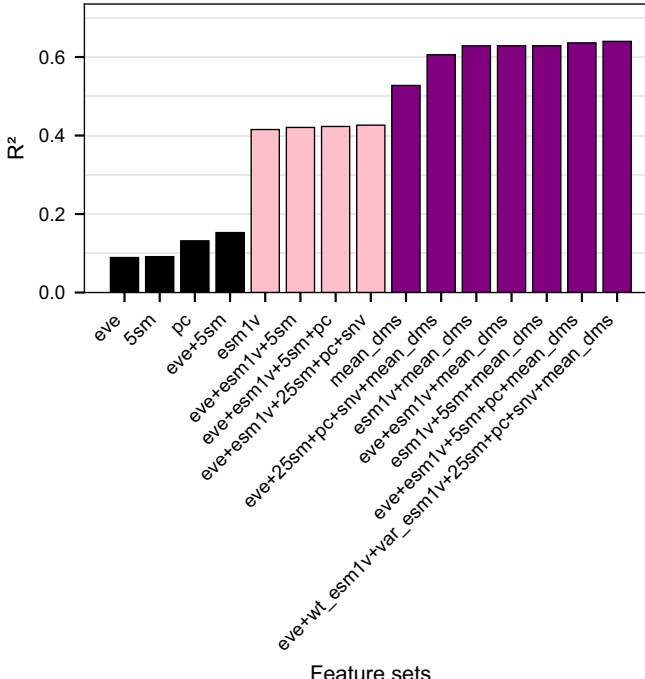

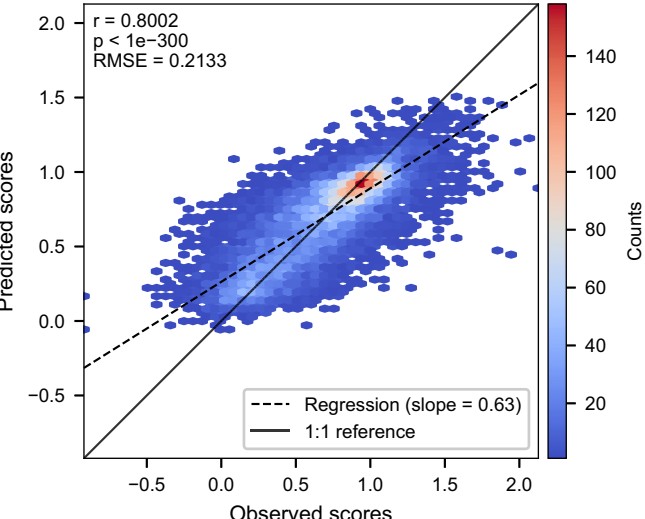

**Figure 2. General cross-protein model performance ($R^2$) for DMS score imputation across feature sets.**

Models were trained on the 140 protein domains for which EVE scores were available. Predictive power increased with feature richness, especially when ESM-1v embeddings and positional priors (mean DMS scores) were included. The top model ($R^2 = 0.64$, $r = 0.80$) excluded only the ESM-1v difference vector (see Dataset EV2 for full definitions of feature sets, and Appendix Fig. S2 and Dataset EV3 for full results). Source data are available online for this figure.

**Figure 3. Correlation between predicted and observed DMS scores for the best general cross-protein model.**

Predicted scores are from the model using all features except the ESM-1v difference embedding. The 2D histogram shows prediction density. Most predictions cluster around the wild-type score of 1, reflecting the prevalence of near-neutral variants. The normalized DMS scores are theoretically bounded between 0 (nonsense-like) and 1 (neutral-like), but both predicted and observed scores occasionally fall outside this range. This is due to the fact that some experimental DMS datasets contain out-of-bound values, which are preserved during normalization. The 1:1 reference line shows perfect prediction ($y = x$). Pearson correlation analysis showed a strong positive association between predicted and observed scores ($r = 0.8002$, $p < 2.2 \times 10^{-308}$, two-sided test). Source data are available online for this figure.

## General cross-protein (LOPO) model evaluation

To further evaluate generalizability, we trained 140 leave-one-protein-out (LOPO) models—each excluding one protein domain for testing. The model achieved consistently strong predictive performance across domains, with median values of $r = 0.82$, $R^2 = 0.66$, RMSE $= 0.20$, and MAE $= 0.15$ (Fig. 4A).

To further highlight domain-specific accuracy, we identified the top 10 best-performing protein domains under the LOPO setup, based on $R^2$ values. These domains have $R^2$ scores between 0.78 and 0.87 (Fig. 4B), indicating highly reliable DMS score imputation for structurally and functionally diverse protein domains.

## Per-protein model evaluation: random splits, LPosO, split by SNV, split by AA class

We evaluated the predictive performance of per-protein models across various data partitioning strategies and feature sets. As expected, models trained on a greater proportion of the data (90%, 85%, and 80%) achieved better performance (Fig. 5A) compared to those trained on more limited subsets (20%, 15%), underscoring the importance of data richness in training (Fig. 5A). To assess robustness under data scarcity, we simulated a high-sparsity scenario by reducing training data to as little as 15%, observing a modest decline in performance. In the most extreme configuration

(15% training/85% test), the average Pearson correlation dropped by up to 0.1 compared to the richer configuration (85% training/15% test) across the top 10 proteins (Fig. 5A). These results highlight the adaptability of VEFill in low-data regimes and underscore its potential for guiding targeted mutagenesis efforts by prioritizing informative variants.

Performance comparison between the random 80/20 split and the leave-position-out (LPosO) configuration for the top 10 proteins—ranked by LPosO performance—revealed consistent trends in predictive accuracy. The LPosO setup, which simulates a more stringent scenario by withholding all variants from 20% of sequence positions during training, demonstrated the model's capacity to generalize across distinct positions of the protein sequence (Fig. 5B).

We also tested how reduced feature sets influenced model performance. Models trained using only ESM-1v and mean DMS scores, as well as models trained without ESM-1v embeddings, retained competitive performance compared to models trained on full feature set (Fig. 5C). The limited contribution of ESM-1v in this per-protein context reflects the fact that, under random 80/20 splits, most positions are already represented in the training data. Consequently, the positional mean DMS score acts as a strong empirical prior that captures much of the site-specific mutational tolerance, leaving little residual variance for contextual sequence embeddings to explain. This stands in contrast to the cross-protein setting (Fig. 2), where positional means are unavailable for new proteins and ESM-1v plays a central role in enabling generalization.

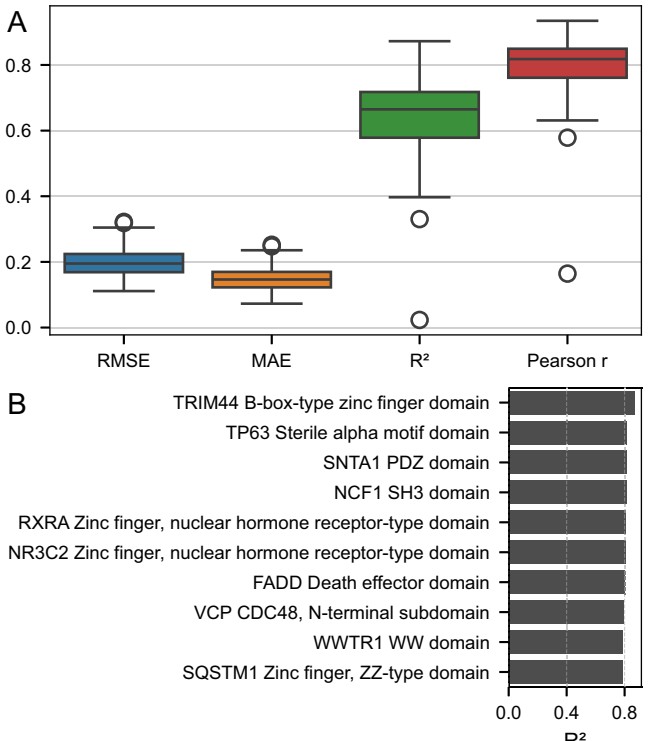

**Figure 4. Performance evaluation of the general cross-protein (LOPO) model across protein domains.**

(A) Evaluation metrics for the general cross-protein (LOPO) model trained and tested across 140 protein domains. Each model was trained using data from 139 proteins and tested on the one held-out protein. Each box summarizes RMSE, MAE, $R^2$, and Pearson correlation coefficient across all runs. The center line indicates the median; the box bounds represent the interquartile range (25th–75th percentile); whiskers extend to the most extreme values within 1.5× the interquartile range from the lower and upper quartiles; and points beyond the whiskers denote outliers. (B) $R^2$ values of the top 10 best-performing proteins from the same LOPO setup. Each bar represents a protein that was excluded from training and used for testing. Source data are available online for this figure.

Moreover, we compared performance between models trained exclusively on single-nucleotide variants (SNVs) and those trained using a 42/58 random train/test split, which reflects the average training proportion observed in the SNV-based splits. Despite training on a more restricted subset of mutations, models trained on SNVs alone achieved comparable performance to those trained on the broader mutation set, indicating that informative SNVs can capture key predictive signals (Fig. 5D).

Finally, we evaluated whether training exclusively on a small set of representative amino acid substitutions can provide sufficient predictive signal for accurate imputation. To test this, we selected five substitutions—histidine, glutamic acid, asparagine, isoleucine, and glycine—that each represent a distinct physicochemical class (positive, negative, polar, hydrophobic, and special, respectively). These specific amino acids were chosen because they are closest to the median substitution within their class in the comprehensive analysis of (Gray et al, 2017). Models trained exclusively on variants containing one of these five substitutions (H, E, N, I, G) were compared against models trained using a random 23/77 train/

test split matched to the average training proportion observed for these subsets. Despite training on a greatly restricted mutation set, models trained only on the (H, E, N, I, G) substitutions achieved performance comparable to the broader 23/77 baseline split (Fig. 5E). This result indicates that a carefully selected set of representative substitutions can capture much of the relevant mutational landscape and could serve as a feasible foundation for designing sparse, information-efficient variant libraries.

### Fine-grained per-protein evaluation: LOPosO vs. LOVarO

To explore model generalization at finer granularity, we compared leave-one-position-out (LOPosO) and leave-one-variant-out (LOVarO) strategies on two top-performing models for FADD Death effector domain and TRIM44 B-box-type zinc finger domain. In the LOPosO setup, all variants at a single position were held out during each iteration, evaluating the model's ability to generalize across positions. LOVarO, in contrast, held out a single variant at a time, enabling localized performance evaluation within familiar sequence contexts.

Both strategies showed consistently low error rates, affirming the model's robustness. However, LOPosO exhibited slightly higher prediction variability, consistent with its broader generalization demands (Fig. 6A,B).

Several mutation-specific patterns emerged from the squared error distributions. Proline (P) substitutions consistently led to elevated prediction errors across both strategies (Fig. EV2), likely due to proline's unique conformational constraints and disruption of secondary structures—effects that are not easily captured by sequence-based features such as ESM-1v embeddings.

We also observed that extremely damaging variants, particularly those with normalized scores below the loss-of-function threshold, were more prone to higher squared error. This occurred despite overall strong performance for most damaging variants, reflecting the inherent challenge of modeling rare, high-impact effects.

Interestingly, for TRIM44 B-box-type zinc finger domain, several histidine-to-cysteine (H → C) mutations were poorly predicted by LOPosO but accurately captured by LOVarO. This discrepancy highlights the value of fine-grained, position-aware evaluation in capturing context-dependent biochemical nuances. Histidine and cysteine, for instance, can coordinate zinc ions in domain-specific roles that LOPosO may misclassify due to its generalized view (Cassandri et al, 2017) (Figs. 6A,B and EV2, see Appendix Fig. S3 for additional insight into correlation between observed scores and prediction error magnitudes).

### Performance evaluation relative to the DMS assay noise ceiling

Because experimental noise imposes a fundamental upper bound on achievable predictive accuracy, we used the noise ceiling estimation procedure to evaluate VEFill's per-protein performance relative to the maximum accuracy permitted by DMS assay variability. Across the 139 domains (after excluding one domain with too few variants), noise ceilings ranged from 0.68 to 0.99, reflecting differences in experimental precision across assays. VEFill's per-protein Pearson correlations ($r = 0.52$–$0.91$ under the 80/20 split) lie consistently below, but generally close to, these ceilings (Appendix Fig. S4). Occasional cases in which VEFill

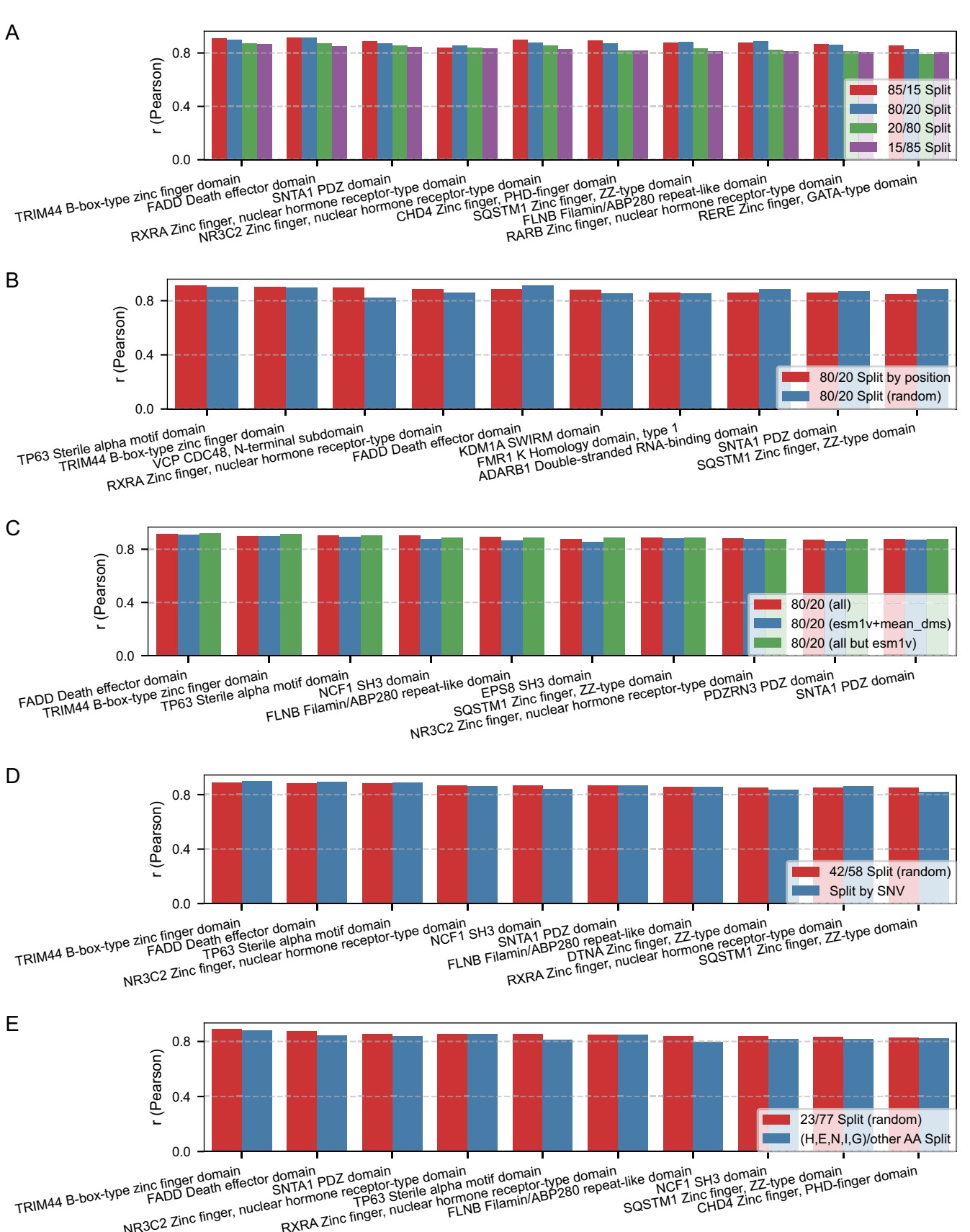

◄ **Figure 5.   Performance of the top 10 per-protein models based on Pearson correlation coefficient ($r$) between observed and predicted DMS scores under various data splitting strategies and feature sets.**

(A) Comparison of per-protein models trained using four different random train/test splits: 85/15, 80/20, 20/80, and 15/85. The y-axis shows the $r$ value for one protein's model, with the top 10 proteins ranked by performance in the 15/85 split. (B) Performance comparison between random 80/20 splitting and the leave-position-out (LPosO) strategy for the top 10 proteins (based on LPosO performance). In LPosO, all variants at 20% of the positions are held out during training. (C) Effect of feature selection on per-protein model performance. All models use the 80/20 random split, and three feature sets are compared: full feature set, ESM-1v embeddings and mean DMS score only, and full set excluding ESM-1v. The top 10 per-protein models are selected based on their performance without ESM-1v features. (D) Comparison of model performance using two training strategies: training exclusively on SNVs, and a 42/58 random train/test split, which corresponds to the average split ratio in the SNV-based models. The top 10 models are selected based on performance with the 42/58 train/test split. (E) Comparison of model performance using two training strategies: training exclusively on five representative amino acid substitutions (His, Glu, Asn, Ile, and Gly), and a 23/77 random train/test split, which corresponds to the average split ratio observed for the (H, E, N, I, G)-based models. The top 10 proteins are selected based on performance with the 23/77 train/test split. Source data are available online for this figure.

slightly exceeds the estimated ceiling are expected due to sampling variability in both the simulated replicates and the correlation estimates. Collectively, these results demonstrate that VEFill performs near the accuracy limit imposed by experimental noise, and that much of the remaining discrepancy between predicted and observed scores is attributable to measurement uncertainty rather than model underfitting.

Although measurement uncertainty imposes an upper limit on achievable accuracy, we additionally evaluated whether filtering out variants with the highest reported assay uncertainty improves model performance. For each protein domain, we progressively removed the top 10%, 20%, and 30% most uncertain variants and retrained the per-protein models. Median Pearson correlations remained effectively unchanged across all filtering levels (Appendix Fig. S5), indicating that VEFill does not substantially benefit from excluding low-confidence measurements. Nonetheless, large discrepancies between VEFill predictions and experimentally reported scores—particularly for variants with high assay uncertainty—may help flag measurements that warrant further experimental verification.

## Benchmarking against existing DMS imputation methods

To assess the practical performance of VEFill relative to other DMS imputation approaches, we compared it against a range of existing methods, including a residue-mean baseline, Wu et al (2019), Envision, FactorizeDMS, AALasso, and two five-nearest-neighbor approaches, across multiple benchmarking regimes.

To enable fair comparison across models with differing feature requirements, benchmarking was performed across three complementary settings: (i) all 146 proteins with features available per method (Fig. EV3A); (ii) a strict 40-protein subset on which all benchmarked models could be applied to exactly the same datasets, enabling direct, like-for-like comparison (Fig. EV3B); and (iii) a 68-protein subset that allowed additional cross-model comparison for all methods except VEFill-max (the full-feature model), which requires EVE scores that were unavailable for part of this set (Fig. EV3C). Across all evaluation settings, VEFill-min (a reduced-feature model using only the positional mean DMS score and ESM-1v embeddings for the wild-type, variant, and their difference) consistently outperformed all other models, while VEFill-max ranked third overall.

To examine how imputation performance depends on dataset completeness, we conducted a controlled subsampling analysis on 28 high-quality DMS datasets with complete feature availability and overall completeness of at least 94.7%, holding out 10% of variants as a

fixed test set and subsampling the remaining variants at increasing fractions (10–90%) to simulate partially complete experiments. Across models, distinct trends emerged as data availability increased (Fig. 7). Some approaches, including Wu et al (2019) and Envision, showed relatively flat performance across completeness levels, whereas FactorizeDMS and AALasso improved gradually as more data became available. In the lowest-data regime (10% training fraction), VEFill-max achieved the highest median Pearson correlation across datasets, while VEFill-min ranked third, slightly below Envision. From 20% completeness onward, both VEFill-min and VEFill-max consistently outperformed all other models. The 5NN-Functional baseline reached comparable performance only after approximately 60% completeness, and from 70% onward its performance became nearly indistinguishable from VEFill.

To further evaluate VEFill's behavior in low-data regimes relevant to sparse library design, we analyzed performance as a function of the number of experimentally measured substitutions per position. Using the same 28 datasets, per-protein VEFill-min and VEFill-max models were trained while restricting the number of mutations per site ($N = 1$–20), with mean DMS scores recomputed dynamically from the available substitutions only. Across datasets, both VEFill-min and VEFill-max showed rapid performance gains as additional substitutions per position became available, with predictive accuracy beginning to plateau at approximately $N \approx 4$ mutations per site (Appendix Fig. S6). Beyond this point, additional measurements yielded diminishing returns. These results indicate that the positional mean DMS score becomes a stable and informative feature once a small number of substitutions are observed at each position, and that VEFill can effectively leverage such sparse positional information in combination with sequence-derived features. Notably, this regime is consistent with the completeness-based analysis (Fig. 7), where VEFill models begin to consistently outperform alternative imputation approaches once approximately 20% of variants are available, corresponding to a similar number of measurements per position. Together, these findings demonstrate that VEFill can support the design and analysis of sparse yet informative variant libraries without requiring full site-saturation mutagenesis for assay readouts like protein stability.

## Completeness of existing DMS datasets and practical motivation for VEFill

To evaluate the practical relevance of VEFill in settings where partial DMS measurements are available, we conducted a

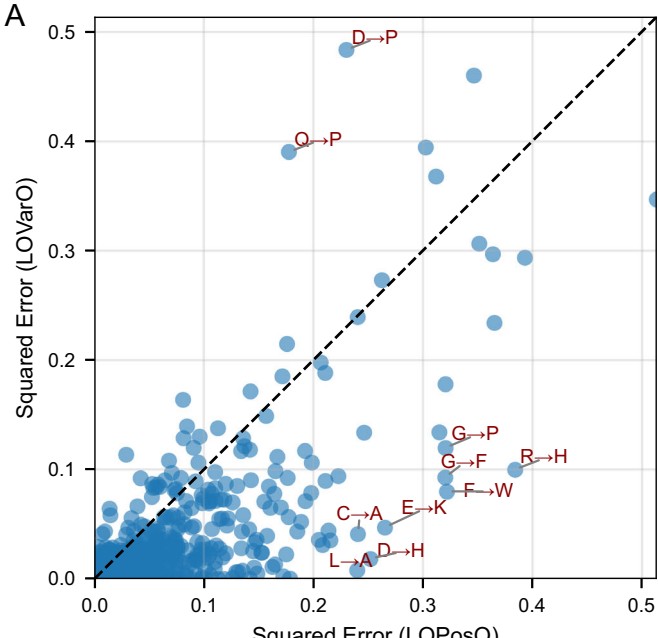

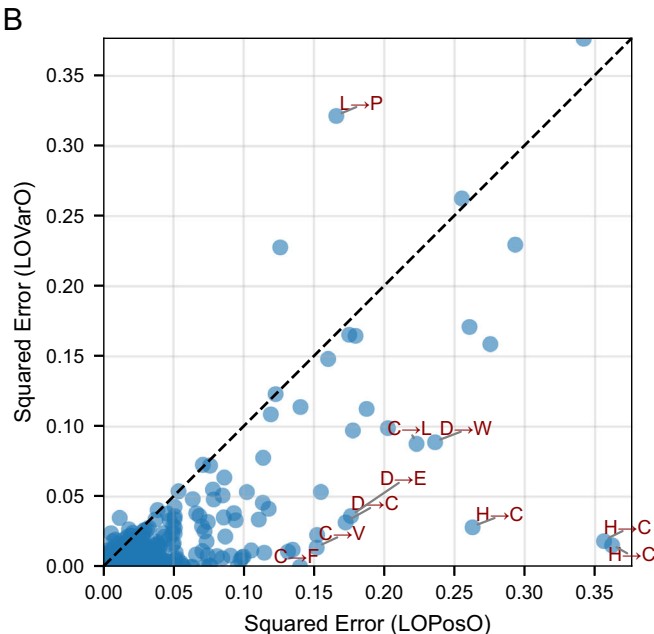

**Figure 6. Comparison of LOPosO and LOVarO fine-grained prediction errors for two representative proteins.**

(A) FADD death effector domain. (B) TRIM44 B-box-type zinc-finger domain. Each scatter plot compares variant-level prediction errors obtained under the leave-one-position-out (LOPosO) and leave-one-variant-out (LOVarO) strategies. Points represent individual single-amino-acid substitutions; the diagonal line indicates identical error between the two strategies. Labels highlight the top 10 substitutions with the largest discrepancies between LOPosO and LOVarO squared error, indicating cases where position-level versus variant-level generalization differs most strongly. Source data are available online for this figure.

completeness analysis inspired by the framework of Fu (2023). Many contemporary DMS experiments report only a subset of all possible single-amino-acid substitutions, resulting in datasets where positional mean DMS scores can be computed but substantial fractions of variants remain unmeasured. Such settings align closely with VEFill's intended use case—augmenting partially complete experimental maps with accurate imputations for the remaining substitutions.

We analyzed 841 public score sets from the most recent MaveDB Zenodo release (MaveDB Contributors, 2025). For each dataset, we quantified completeness using a modified version of Fu's formulation, defining the total number of possible substitutions at each scanned residue as 20 (the 20 standard amino acids including the *Ter* substitution, minus the wild-type residue). Completeness was calculated as:

$$\text{Completeness} = \frac{\text{Number of reported single-amino-acid variants}}{\text{Number of scanned residues} \times 20}.$$

(1)

Each dataset was then assigned to one of eleven completeness bins spanning the interval [0, 1] in increments of 0.1. This analysis revealed that partially complete datasets are common, with many experiments covering only a subset of possible substitutions, while fully complete datasets are rare, with only 44 out of 841 achieving complete coverage (Dataset EV4).

Importantly, our benchmarking results (Figs. 7 and EV3) demonstrate that VEFill begins to outperform alternative imputation strategies once at least 20% of variants have been experimentally measured (i.e., completeness ≥0.2). When combined with the completeness distribution observed across MaveDB, this indicates that a large proportion of real-world DMS datasets fall squarely within the regime where VEFill provides the most substantial benefit (Table 1).

## Generalization to non-Domainome assays

The general cross-protein model trained on the Human Domainome 1 dataset was evaluated on 8 full-length, non-Domainome proteins from MaveDB that were not seen during training. The model showed substantially better performance on stability-based assays (Clausen et al, 2024; Grønbæk-Thygesen et al, 2024; Matreyek et al, 2021) than on activity-based ones (Giacomelli et al, 2018; Kotler et al, 2018; Mighell et al, 2018; Weile et al, 2017) (Table 2, Fig. 8A,B; Appendix Figs. S7 and S8). This demonstrates the importance of generating experimental data for diverse assay types to enable more powerful computational models.

## Minimal feature model for imputation

We tested a lightweight version of the cross-protein model that uses (i) only ESM-1v embeddings (wild-type, variant, and difference vectors) combined with mean DMS scores per position—two features identified as highly important in our feature selection analysis—and (ii) mean DMS scores alone. Reduced-feature models were trained on either 140 or 521 protein domains from the Human Domainome 1 dataset.

We compared the performance of these models with the full-feature model using a validation set (10% holdout from Human Domainome 1) and external evaluation on full-length non-

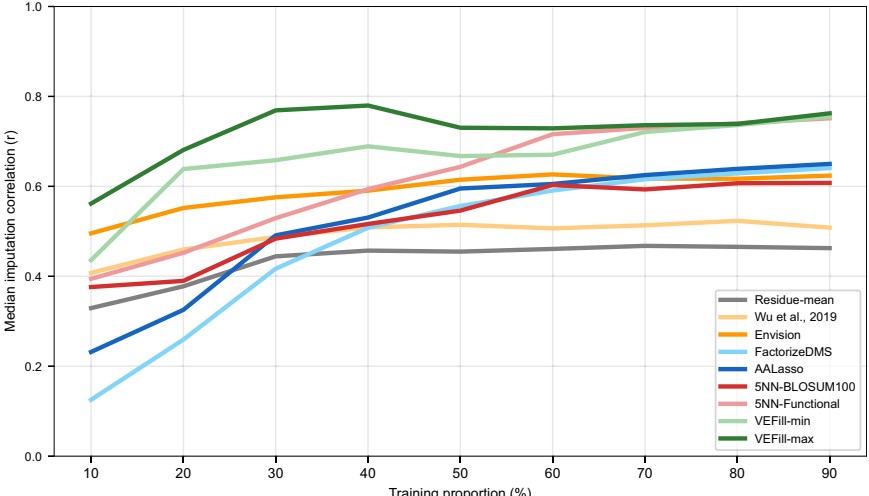

**Figure 7. Impact of dataset completeness on DMS imputation accuracy across models.**

Imputation performance as a function of available training data for 28 high-quality DMS datasets. For each dataset, 10% of variants were held out as a fixed test set, while the remaining variants were subsampled at increasing fractions (10–90% of the full dataset) to simulate incomplete experiments. Each curve shows the median Pearson correlation between predicted and experimental DMS scores across all datasets for a given model. VEFill uses dynamically recomputed positional mean DMS scores based only on the available training subset at each completeness level. Source data are available online for this figure.

**Table 1. Completeness distribution across 841 contemporary MAVE/DMS datasets and representative examples from each bin.**

| Completeness bin | Datasets in bin (*n*) | Proportion (*n*/841) | Example protein | Example completeness |
|---|---|---|---|---|
| $0.2 \leq c < 0.3$ | 15 | 0.0178 | VKOR (Chiasson et al, 2020; Data ref: Chiasson et al, 2020) | 0.2438 |
| $0.3 \leq c < 0.4$ | 15 | 0.0178 | BRCA1 (Adamovich et al, 2022; Data ref: Adamovich et al, 2022) | 0.3736 |
| $0.4 \leq c < 0.5$ | 6 | 0.0071 | TEM-19 β-lactamase (Firnberg et al, 2014; Data ref: Firnberg et al, 2014) | 0.4809 |
| $0.5 \leq c < 0.6$ | 8 | 0.0095 | PTEN (Matreyek et al, 2021; Data ref: Matreyek et al, 2021) | 0.5971 |
| $0.6 \leq c < 0.7$ | 15 | 0.0178 | PTEN (Matreyek et al, 2018; Data ref: Matreyek et al, 2018) | 0.6105 |
| $0.7 \leq c < 0.8$ | 29 | 0.0345 | INSR (Aslanzadeh et al, 2025; Data ref: Aslanzadeh et al, 2024) | 0.7292 |
| $0.8 \leq c < 0.9$ | 140 | 0.1665 | KRAS (Weng et al, 2024; Data ref: Weng et al, 2024) | 0.8936 |
| $0.9 \leq c < 1.0$ | 525 | 0.6243 | STIM1 (Kamath and Matreyek, 2025; Data ref: Kamath and Matreyek, 2025) | 0.9847 |

Completeness was computed as the number of reported single-amino-acid variants divided by the total number of possible substitutions. The table summarizes the number and proportion of datasets falling into each completeness interval ≥0.2, and provides one representative example dataset per bin. These examples illustrate realistic cases where positional mean DMS scores are available but substantial fractions of substitutions remain unmeasured–the regime in which VEFill provides the greatest benefit according to our benchmarking results.

Domainome proteins (PRKN (Clausen et al, 2024), multiple PTEN (Matreyek et al, 2021; Mighell et al, 2018), ASPA (Grønbæk-Thygesen et al, 2024), CALM1 (Weile et al, 2017), TPK1 (Weile et al, 2017), and multiple TP53 (Giacomelli et al, 2018; Kotler et al, 2018)) with 30% of mutations masked, covering both stability- and activity-based DMS datasets (Fig. 9; Appendix Fig. S9). The full-feature model consistently outperformed the reduced versions. However, the reduced models still performed competitively, making them a practical and scalable alternative when only limited features are available.

On the internal test set, the ESM-1v + mean DMS model consistently outperformed the mean DMS-only version. However, this advantage did not consistently translate to the full-length protein dataset, where both reduced models performed comparably across most non-Domainome proteins. Similarly, increasing the

training set size from 140 to 521 domains did not yield substantial changes in performance.

To better understand these results, we analyzed the representation of protein families across the training and test sets of the 140- and 521-domain datasets using the Pfam database (Paysan-Lafosse et al, 2025) (Fig. EV4). We observed a skewed distribution where a few dominant Pfam families (e.g., PF00046, PF00018) contributed a large fraction of total protein domains. In the 521-domain dataset, the top 5 Pfam families alone accounted for over 30% of all protein domains. Such overrepresentation introduces redundancy, potentially explaining why increasing the training size from 140 to 521 domains did not yield major improvements in external generalization.

To directly test the effect of this redundancy, we constructed a reduced dataset from Human Domainome 1 containing one protein

**Table 2. General cross-protein model performance on external assays.**

| Protein | Assay | RMSE | R² |
|---|---|---|---|
| PRKN (Clausen et al, 2024; Data ref: Clausen et al, 2024) | Stability | 0.175 | 0.799 |
| PTEN (Matreyek et al, 2021; Data ref: Matreyek et al, 2021) | Stability | 0.214 | 0.550 |
| ASPA (Grønbæk-Thygesen et al, 2024; Data ref: Grønbæk-Thygesen et al, 2024) | Stability | 0.229 | 0.702 |
| CALM1 (Weile et al, 2017; Data ref: Weile et al, 2017) | Activity | 0.457 | 0.152 |
| PTEN (Mighell et al, 2018; Data ref: Mighell et al, 2018) | Activity | 0.495 | 0.465 |
| TPK1 (Weile et al, 2017; Data ref: Weile et al, 2017) | Activity | 0.544 | 0.158 |
| TP53 (Giacomelli et al, 2018; Data ref: Giacomelli et al, 2018) | Activity | 0.793 | 0.473 |
| TP53 (Kotler et al, 2018; Data ref: Kotler et al, 2018) | Activity | 1.438 | −0.125 |
| Human Domainome 1 (Beltran et al, 2025) | Stability | 0.213 | 0.640 |

Summary of general cross-protein model performance tested on external stability- and activity-based assays. The final row (Human Domainome 1) presents the model's performance on internal validation data and serves as a reference benchmark for assessing generalization to unseen proteins.

domain per Pfam family (127 domains in total, see Dataset EV1 for the full list of domains and their inclusion in the 140-, 521-, and 127-domain (Pfam-unique) datasets), ensuring complete non-overlap of Pfam families between training and test sets. Models trained on this Pfam-unique set exhibited slightly higher RMSE compared to those trained on the full 521-domain set: 0.2654 vs. 0.2039 for the ESM-1v + mean DMS model, and 0.3671 vs. 0.3012 for the mean DMS-only model. These modest increases suggest that while Pfam redundancy exists, it does not strongly bias performance or lead to overfitting in our reduced-feature models.

## Zero-shot generalization (no mean DMS)

To assess the model's potential for zero-shot prediction, we evaluated a general cross-protein model trained without the positional mean DMS score feature on external, non-Domainome datasets. Since the mean DMS score per position is derived directly from experimental data, removing this feature ensures no data leakage from training to testing—a necessary condition for genuine zero-shot evaluation.

We tested the model on both stability-based (PRKN, PTEN, ASPA) and activity-based (CALM1, PTEN, TPK1, TP53) external assays. While performance decreased compared to the full-feature version (Table 2, Fig. 8A,B; Appendix Figs. S7 and S8), especially on activity-based assays, the model still demonstrated relatively better accuracy on stability-based over activity-based datasets (Fig. 10; Appendix Figs. S10 and S11). These results confirm that although the model can still capture useful general patterns, its accuracy without access to local experimental context (mean DMS scores) remains limited.

## Discussion

Our findings demonstrate that VEFill—a LightGBM-based model integrating evolutionary, biochemical, and sequence-based features —enables accurate imputation of deep mutational scanning (DMS) scores across a diverse range of human protein domains. Trained on the standardized and stability-focused Human Domainome 1 dataset, VEFill achieved strong predictive performance, with the highest gains observed when using contextual embeddings (ESM-1v) and position-specific priors (mean DMS score per position).

These results underscore the utility of combining data-driven positional information and learned sequence representations to enhance model generalization.

Evaluation on both internal and external datasets confirmed VEFill's capacity to generalize across previously unseen domains assayed under stability-focused experimental conditions. However, its performance on activity-based datasets was markedly reduced, likely due to mismatches between training and test assay modalities, as well as the complex nature of cellular phenotypes. A systematic evaluation across a curated subset of compatible MaveDB datasets would be valuable for further characterizing model performance across assay types, but would require careful assay harmonization.

Performance assessments of VEFill under various per-protein train-test configurations (random splits, LPosO, LOPosO, LOVarO) showed strong robustness in sparse-data scenarios. LOVarO consistently outperformed other strategies, particularly for context-specific or chemically subtle mutations, underscoring the importance of maintaining positional information during training. Notably, substitutions involving proline residues posed consistent prediction challenges, likely due to their disruptive conformational effects not fully captured by the feature set. Additionally, extremely deleterious variants—especially those falling below the normalized loss-of-function threshold—tended to have higher prediction error, despite overall strong model performance on damaging mutations. This reflects the inherent difficulty of modeling rare, high-impact outliers. Specific mutation types, such as histidine-to-cysteine substitutions in zinc finger domains, were predicted more accurately under fine-grained LOVarO validation compared to LOPosO, further emphasizing the importance of preserving positional and structural nuances.

Across multiple benchmarking analyses, per-protein VEFill demonstrates robust and competitive performance relative to a broad range of existing DMS imputation methods, including residue-wise mean baselines, Wu et al (2019), Envision, Factor-izeDMS, AALasso, and two five-nearest-neighbor approaches. In controlled completeness-based benchmarking, two different VEFill settings consistently outperform all previously published models once at least 20% of variants have been experimentally measured. Importantly, noise-ceiling analyses indicate that per-protein VEFill operates near the accuracy limit imposed by experimental variability in DMS assays, suggesting that further performance

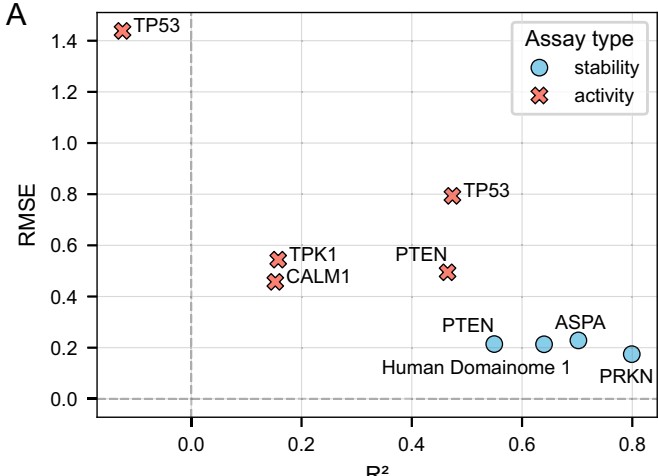

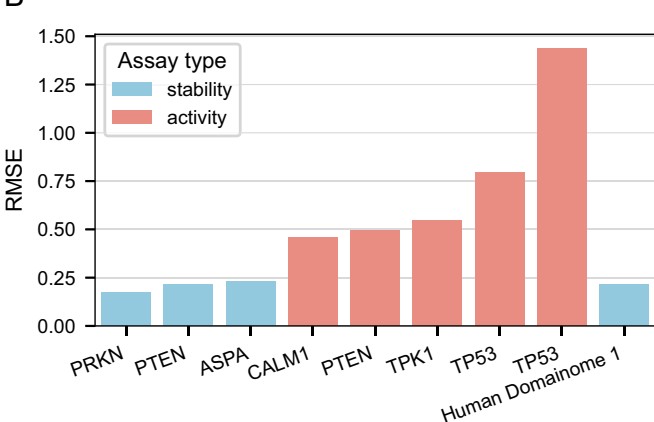

**Figure 8. Performance of the general cross-protein model across stability-based and activity-based external assays.**

(A) Comparison of RMSE and $R^2$ values across assays, illustrating substantially better predictive accuracy on stability-based assays (PRKN, PTEN, ASPA) compared to activity-based assays (CALM1, PTEN, TPK1, TP53). (B) Explicit comparison of RMSE differences, highlighting the notable increase in prediction error for activity-based assays relative to stability-based assays. Performance on internal validation (Human Domainome 1) data is included as a reference. See Appendix Fig. S7(A–C) for additional performance metric visualizations and Appendix Fig. S8(A–H) for correlations of predicted versus observed DMS scores for each assay. Source data are available online for this figure.

per site. Collectively, this body of evidence supports per-protein VEFill as a practical framework for augmenting partial DMS datasets and guiding efficient experimental design under realistic data constraints.

A reduced version of VEFill, incorporating only ESM-1v embeddings and mean DMS scores, achieved performance comparable to the full model in most cases. This lightweight configuration may be particularly useful in resource-constrained settings. However, excluding position-level priors significantly reduced accuracy in zero-shot scenarios, highlighting their critical role in enabling generalizable variant effect prediction.

To further investigate model generalization, we evaluated the performance of reduced-feature models trained on either 140 or 521 protein domains and analyzed the distribution of Pfam families across these training datasets. A highly skewed distribution was observed, where a small number of Pfam families dominated the dataset. In the 521-domain set, for instance, the top five Pfam families alone accounted for over 30% of all protein domains. This biological redundancy reflects true enrichment of certain protein families in the human genome, but it also limits family diversity. As a result, increasing the training size from 140 to 521 domains mainly added examples from already well-represented families, offering limited coverage of novel Pfam families. Consequently, the larger dataset did not confer clear improvements in external generalization, particularly when test proteins belonged to families absent from the training data.

We emphasize that protein family overlap between training and test sets is not a data leakage artifact but a biologically relevant feature of protein space. Domains within the same Pfam family tend to share structural and functional properties, and generalizing within families is desirable. To explicitly test whether shared Pfam families may inflate performance, we repeated training using only one protein domain per Pfam family (127 total), thereby eliminating any Pfam-level leakage between train and test splits. RMSE increased only modestly (by 0.07 for both reduced-feature models), suggesting that while redundancy contributes to model learning, it is not a potential source of overfitting. These results support the robustness of our findings and reinforce that increasing Pfam diversity, rather than dataset size alone, may be more important for generalization across structurally unrelated proteins.

VEFill offers a scalable and interpretable framework for DMS score imputation. Its effectiveness in low-data regimes suggests utility in guiding experimental design, prioritizing mutations for targeted assays, and constructing efficient mutational libraries. Beyond current capabilities, the model could be further improved by explicitly incorporating structural features. Although VEFill does not directly use protein structure, ESM-1v embeddings may implicitly encode structural patterns. Nonetheless, we observed a decline in predictive performance at structurally sensitive positions —such as proline substitutions—indicating that the explicit inclusion of structural features may further improve model accuracy.

Future directions include incorporating structural modeling predictions or annotations to address context-specific error patterns. Cross-species transfer learning, leveraging multiple sequence alignments, may also extend VEFill's utility to predict variant effects in non-human proteins using experimental data from related species.

gains are primarily constrained by measurement uncertainty rather than model capacity.

We find that training per-protein VEFill models using a small, representative set of amino-acid substitutions representing distinct physicochemical classes—specifically histidine, glutamic acid, asparagine, isoleucine, and glycine (H, E, N, I, G)—yields predictive performance comparable to that obtained using randomly selected substitutions at matched training coverage. Together with the observation that per-protein VEFill performance begins to plateau after approximately four mutations per position, these findings suggest that informative yet non-exhaustive variant libraries can be designed by strategically selecting a limited number of substitutions

# Methods

## Reagents and tools table

| Reagent/ Resource | Reference or Source | Identifier or Catalog Number |
|---|---|---|
| Experimental models | | |
| Recombinant DNA | | |
| Antibodies | | |
| Oligonucleotides and other sequence-based reagents | | |
| Chemicals, Enzymes and other reagents | | |
| Software | | |
| VEFill v1.0.1 | https://github.com/PlushZ/VEFill | |
| Python v3.10 | Python Software Foundation | |
| LightGBM v4.5.0 | https://github.com/microsoft/LightGBM Ke et al, 2017 | |
| Scikit-learn v1.6.0 | https://scikit-learn.org/ Pedregosa et al, 2011 | |
| Optuna v4.1.0 | https://optuna.org/ Akiba et al, 2019 | |
| Pandas v2.2.3 | https://pandas.pydata.org/ The pandas development team, 2024 | |
| NumPy v2.2.0 | https://numpy.org/ Harris et al, 2020 | |
| SciPy v1.14.1 | https://scipy.org/ Virtanen et al, 2020 | |
| Matplotlib v3.10.7 | https://matplotlib.org/ Hunter, 2007 | |
| seaborn v0.13.2 | https://seaborn.pydata.org/ Waskom, 2021 | |
| Biopython v1.83 | https://github.com/biopython/biopython Cock et al, 2009 | |
| PostgreSQL v14.17 | https://www.postgresql.org/ PostgreSQL Global Development Group | |
| fair-esm v2.0.0 | https://github.com/facebookresearch/esm Meier et al, 2021 | |
| EVE | https://evemodel.org/ Frazer et al, 2021 | |
| MaveDB | https://www.mavedb.org/ Rubin et al, 2025 | |
| Other | | |

## Data collection and storage

For this study, we collected DMS datasets from MaveDB, focusing on the Human Domainome 1 dataset, which contains mutagenesis data for 522 protein domains. One of three artificial (non-natural) sequences from the original Human Domainome 1 dataset was excluded due to its absence in MaveDB, resulting in 521 domains available for downstream analysis. This decision ensured consistency in data sourcing and prioritized biologically grounded sequences.

For general cross-protein model development, we used a filtered subset of 140 domains for which EVE scores were available. This filtering step ensured consistent feature representation across all proteins and avoided introducing data sparsity or inconsistencies in downstream modeling. The set of all 521 domains was used to train reduced-feature models relying only on ESM-1v embeddings and mean DMS scores, simulating settings where fewer feature types are available (see Dataset EV1 for inclusion details).

EVE scores were retrieved from the official EVE model website. A total of 25 amino acid substitution matrices were obtained—24 from the Biopython v1.83 package (Cock et al, 2009) (via Bio.Align.substitution_matrices) and one from a publicly available source (Appendix Table S1). Physicochemical properties were derived from established biochemical literature and databases. Features such as hydrophobicity, solvent accessibility, charge, chemical group classifications, molecular weight, pKa values, isoelectric point, hydropathy index, and others were obtained from curated biochemical sources, including the AAindex database (Kawashima and Kanehisa, 2000), Kyte-Doolittle hydropathy scale (Kyte and Doolittle, 1982), and Lehninger Principles of Biochemistry (Nelson and Cox, 2017) (see Dataset EV2 for the full list of features used for VEFill, including physicochemical properties).

Protein embeddings were generated using the protein language model ESM-1v (650M parameters, UR90S/1 variant; model identifier: esm1v_t33_650M_UR90S_1.pt) (Meier et al, 2021). The pre-trained model weights were downloaded from the FAIR ESM repository (https://github.com/facebookresearch/esm). For each protein, we retrieved residue-level embeddings for both the wild-type sequence and corresponding single-amino-acid variants from the model's final layer (layer 33), utilizing the official fair-esm Python package (version 2.0.0). Additionally, we computed difference vectors between wild-type and variant embeddings at the mutated positions to specifically capture localized sequence changes.

To reconcile different indexing schemes—MaveDB reports positions relative to the domain (1-based), whereas EVE uses UniProt (UniProt Consortium, 2025) full-length coordinates—we identified offset values for each protein to allow for correct mapping during preprocessing.

All collected data, including target amino acid sequences, UniProt offsets, DMS scores, EVE scores, substitution matrices, embeddings, and other features, were stored in a PostgreSQL v14.17 database for structured, scalable, and unified access throughout the modeling pipeline (see Appendix Fig. S1 for a schema of the database architecture).

All figures used in this manuscript were generated using Matplotlib v3.10.7 (Hunter, 2007) and seaborn v0.13.2 (Waskom, 2021), and statistical analyses were performed using SciPy v1.14.1 (Virtanen et al, 2020).

## Feature selection

To capture the multifactorial nature of mutational effects, we integrated diverse feature types into our model. These were selected to reflect evolutionary constraints, biochemical properties, and sequence context.

Substitution matrices (e.g., BLOSUM62, PAM250 (Dayhoff et al, 1978)) provide general evolutionary-informed scores for amino

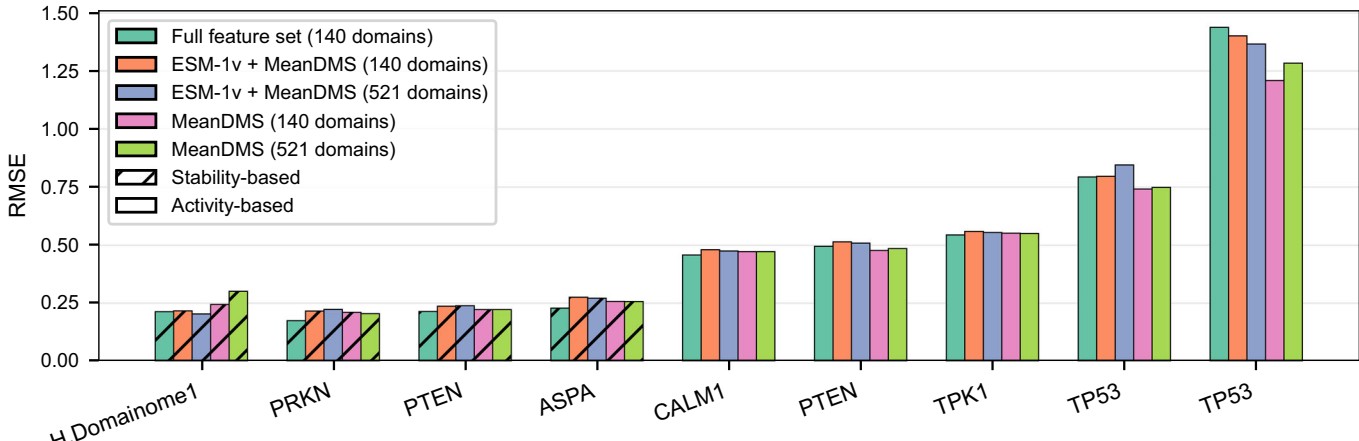

**Figure 9.  Performance comparison of full-feature and reduced-feature models using RMSE as the primary evaluation metric.**

The full-feature model includes all input features, while reduced models include either only ESM-1v embeddings and mean DMS scores or solely mean DMS scores. All reduced-feature models were trained on either 140 or 521 protein domains from the Human Domainome 1 dataset. RMSE values are shown for internal validation on the Human Domainome 1 test set and for external full-length proteins assayed across multiple platforms. Patterned bars denote stability-based assays, which consistently yield stronger predictive performance. The full-feature model outperforms the reduced models across nearly all datasets. Source data are available online for this figure.

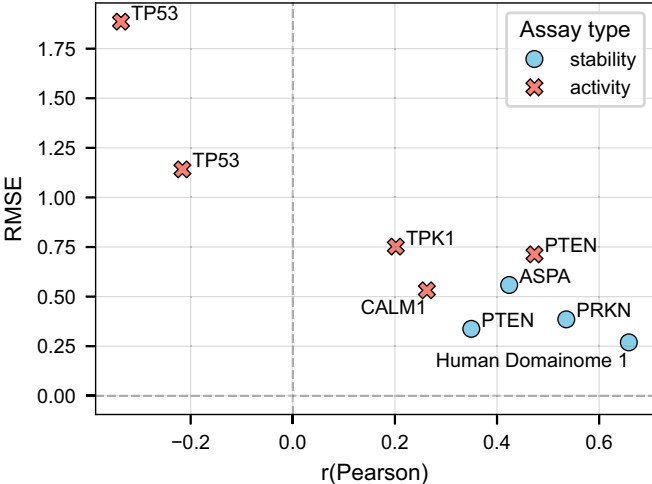

**Figure 10.  Zero-shot general cross-protein model performance without positional mean DMS scores.**

RMSE versus Pearson correlation (*r*) of zero-shot predictions by the general cross-protein model trained without positional mean DMS scores. Stability-based assays consistently exhibit better predictive accuracy than activity-based assays. The internal Human Domainome 1 validation is shown for reference. See Appendix Fig. S10(A–C) for additional metric comparisons and Appendix Fig. S11(A–H) for correlations of predicted versus observed DMS scores for each assay. Source data are available online for this figure.

acid replacements. Physicochemical properties (e.g., charge, polarity, volume, hydrophobicity) were included for wild-type, variant, and difference values to reflect structural and functional impacts. A single-nucleotide variant (SNV) indicator feature was included to distinguish more naturally likely substitutions from multi-nucleotide changes.

We incorporated ESM-1v embeddings to capture contextual, learned representations of protein sequences, and EVE scores to reflect evolutionary conservation and summarize the information from multiple sequence alignments. Additionally, we included the mean DMS score per position as a data-driven prior to account for site-specific mutational tolerance.

This feature set was designed to balance biological interpretability with high predictive performance (see Dataset EV2 for a complete list of features).

## Model architecture

We selected Light Gradient Boosting Machine (LightGBM) (Ke et al, 2017), a framework optimized for efficiency and scalability, as the core model for VEFill. The task involves structured, high-dimensional tabular data and requires regression-based modeling, making tree-based ensembles particularly suitable. LightGBM offers strong predictive performance on medium-to-large datasets without the need for deep learning-scale data volumes. Its built-in regularization and leaf-wise tree growth help prevent overfitting, while support for GPU acceleration ensures computational efficiency. Compared to fully connected neural networks (classic multilayer perceptrons) (Rumelhart et al, 1986) or transformer-based models (Vaswani et al, 2017), LightGBM achieved competitive accuracy in our preliminary tests with substantially faster training and easier interpretability, making it a pragmatic and effective choice for this study (Appendix Table S2). Model training and evaluation were performed in Python using `LightGBM v4.5.0` and `scikit-learn v1.6.0` (Pedregosa et al, 2011).

## Data normalization and preprocessing

Raw DMS scores were normalized using Eq. (2), where 0 corresponds to the median score of nonsense mutations (loss-of-

function baseline) and 1 corresponds to the wild-type score. The normalized DMS score $s_{norm}$ was computed as:

$$s_{norm} = \left( \frac{s_{dms} - s_{wt}}{s_{wt} - s_{nons}} \right) + 1, \tag{2}$$

where $s_{dms}$ is the raw DMS score of the variant, $s_{wt}$ is the wild-type (WT) score determined by the DMS study, and $s_{nons}$ is the mean score of nonsense mutations determined by the DMS study.

This transformation centers neutral mutations (similar to WT) around 1, while deleterious variants typically fall near 0. Some variants—particularly hyperstable mutations or scoring artifacts—can yield normalized scores slightly above 1 or below 0, reflecting that normalization preserves the relative scale of raw measurements and does not bound scores strictly to the [0, 1] interval.

All features were preprocessed to ensure compatibility with the LightGBM model. Categorical variables—such as amino acid class descriptors (e.g., chemical group, charge)—were one-hot encoded and included for wild-type and variant amino acids. Boolean biochemical descriptors (e.g., hydrogen bonding capability, solvent accessibility) were converted to binary integers (0 or 1). Both boolean (converted to binary) and numerical properties of amino acids (e.g., molecular weight, isoelectric point, pKa, hydropathy index) were included for the wild-type and variant residues, as well as their absolute difference:

$$\Delta p = \left| p_{wt} - p_{var} \right|, \tag{3}$$

where $p$ denotes the respective amino acid property, and $p_{wt}$ and $p_{var}$ are the values of the property for the wild-type and variant amino acids, respectively.

ESM-1v embeddings for wild-type, variant, and difference vectors were flattened into fixed-length numeric arrays and appended as individual input features. The difference vector $\Delta \mathbf{e}$ was computed as:

$$\Delta \mathbf{e} = \mathbf{e}_{var} - \mathbf{e}_{wt}, \tag{4}$$

where $\mathbf{e}_{wt}$ and $\mathbf{e}_{var}$ denote the embedding vectors at the mutated position in the wild-type and variant sequences, respectively. This operation captures the local contextual shift in the learned representation space induced by the mutation.

Additionally, we computed the mean normalized DMS score per residue position, grouped by protein and position index. This feature served as a data-driven prior reflecting mutational tolerance at each site, and was used as an additional input feature. The position-wise mean was computed as:

$$\bar{s}_{P,pos} = \frac{1}{N_{P,pos}} \sum_{i=1}^{N_{P,pos}} s_{norm}^{(i)}, \tag{5}$$

where $\bar{s}_{P,pos}$ is the mean normalized DMS score for protein $P$ at position $pos$, $s_{norm}^{(i)}$ is the normalized score of the $i$th variant at that position, and $N_{P,pos}$ is the total number of variants observed at position $pos$ in protein $P$.

All preprocessing steps were executed using a Python-based pipeline that integrated SQL querying from a `PostgreSQL v14.17` database, followed by transformation using `pandas v2.2.3` (McKinney, 2010) and `NumPy v2.2.0` (Harris et al, 2020). This ensured reproducibility and consistent data formatting across the dataset.

## Hyperparameter optimization

To optimize model performance, we performed hyperparameter tuning using the `Optuna v4.1.0` framework (Akiba et al, 2019), which applies Bayesian optimization through a Tree-structured Parzen Estimator (TPE) sampler (Bergstra et al, 2011). A group-aware 5-fold cross-validation strategy was used, with protein domains (identified by gene IDs from our custom PostgreSQL database) serving as grouping variables to prevent data leakage. The objective was to minimize the root mean squared error (RMSE) across folds.

The LightGBM model was trained using gradient boosting decision trees with up to 1000 boosting rounds and early stopping triggered after 50 consecutive rounds without improvement. The search space covered key hyperparameters including learning rate, maximum depth, number of leaves, subsampling ratios, and regularization strengths (L1 and L2 penalties). Once the optimal configuration was identified, a final model was trained using the selected parameters and evaluated on a held-out 10% test set, stratified by gene ID. Evaluation metrics included RMSE, mean absolute error (MAE), and the coefficient of determination ($R^2$).

## Model variants and training strategies

To evaluate imputation strategies for DMS scores, we trained several variants of LightGBM-based models using different data partitioning schemes and feature compositions. All models shared the common LightGBM architecture and were trained using optimized hyperparameters.

All models were evaluated using RMSE, MAE, $R^2$, and Pearson correlation (r), with group-aware train/test splitting based on gene IDs to prevent data leakage.

### General cross-protein model

The general cross-protein model was trained on 136,854 mutations spanning 140 protein domains from the Human Domainome 1 dataset. A leave-protein-out split was applied, where 90% of the protein domains were used for training and the remaining 10% were held out as a test set. To investigate the contribution of different types of biological information, we trained the general cross-protein model using a range of feature set configurations combining evolutionary (EVE scores, substitution matrices), biochemical (amino acid properties), sequence-derived contextual (ESM-1v embeddings), and position-aware (mean DMS score per position, SNV indicator) features. These configurations are summarized in Fig. 2 and detailed in Appendix Fig. S2 and Dataset EV3, where model performance metrics ($R^2$, Pearson $r$, RMSE, and MAE) are compared across feature combinations. We observed major performance gains upon inclusion of ESM-1v embeddings, with further improvement when mean DMS scores per position were added.

### General cross-protein (LOPO) model

For a more systematic evaluation of model generalizability, we implemented a leave-one-protein-out (LOPO) cross-validation strategy. In this setup, 140 models were trained, each time holding

out one protein domain as the test set while training on the remaining 139 domains.

### Per-protein models

To benchmark localized prediction performance, we trained per-protein models individually for each of the 140 protein domains using several train/test partitioning strategies. In the random split setup, each protein's mutation dataset was randomly divided into training and test subsets using various ratios, including 90/10, 85/15, 80/20, 20/80, and 15/85. In the leave-position-out (LPosO) configuration, positions within each protein were split into training and test sets based on 80/20 and 20/80 ratios, allowing the model to generalize across sequence regions. The leave-one-position-out (LOPosO) approach involved holding out all variants at a single position as the test set while training on variants from the remaining positions. Lastly, in the leave-one-variant-out (LOVarO) strategy, each variant was sequentially held out for testing, with all other variants in the same protein used for training.

In addition to using the full feature set, per-protein models were also trained with reduced feature configurations informed by feature importance analyses from the general cross-protein model. These included a reduced set containing only ESM-1v embeddings and the mean DMS score per position, as well as a set containing all features except ESM-1v embeddings.

### Minimal feature model

To evaluate the impact of training dataset size and feature sparsity, we also trained models on a broader set of 521 protein domains (with overall 562,208 mutations) from the Human Domainome 1 dataset. This larger dataset was used specifically for reduced-feature models containing only ESM-1v embeddings and mean DMS scores. One of the original 522 domains was excluded from this set due to its absence in MaveDB. This setup enabled investigation of model scalability and performance in scenarios where full feature availability was limited. (See Dataset EV1 for the full list of domains and their inclusion in the 140- and 521-domain datasets.)

### Estimation of the DMS assay noise ceiling

Deep mutational scanning measurements contain inherent experimental variability, which imposes an upper bound on the predictive accuracy achievable by any computational model. To quantify this limit, we estimated a noise ceiling for each DMS dataset based on the assay-reported measurement uncertainty (e.g., per-variant standard errors or confidence estimates). The objective of this analysis is to determine the maximum correlation that could be expected between two repeated measurements of the same experiment and thereby define the upper limit against which VEFill's performance should be interpreted.

Let $s_i$ denote the reported normalized DMS score for variant $i$, and let $\sigma_i$ denote its corresponding measurement uncertainty. For each protein domain, we simulated $B = 300$ replicate experiments by adding heteroscedastic Gaussian noise to each variant according to its reported uncertainty:

$$\varepsilon_i^{(b)} \sim \mathcal{N}(0, \sigma_i^2), \tag{6}$$

$$s_i^{(b)} = s_i + \varepsilon_i^{(b)}, \, b = 1, \dots, B. \tag{7}$$

For each simulated replicate dataset, we computed the Pearson correlation between the original measurements and the noisy replicate:

$$r^{(b)} = \mathrm{corr}\left(s, s^{(b)}\right). \tag{8}$$

The distribution $\{r^{(b)}\}_{b=1}^{B}$ reflects the expected agreement between two independent realizations of the same experimental assay under its measured noise characteristics. We define the noise ceiling for a given protein domain as the mean of this distribution.

This procedure was applied to 139 of the 140 feature-complete Human Domainome 1 protein domains. One domain (LIMS2 zinc finger, LIM-type domain) contained too few variants in its test set under the VEFill 80/20 random split to support a stable Pearson correlation estimate and was therefore excluded from this analysis.

### Benchmarking framework and comparative evaluation setup

To enable a direct and reproducible comparison with existing DMS imputation approaches, we adopted and extended the benchmarking framework developed by Fu (2023). Specifically, the selection of the 146 DMS datasets and the 28 high-quality datasets used for benchmarking, the implementations of the Residue-mean baseline, Wu et al (2019), Envision (Gray et al, 2018), FactorizeDMS (Fu, 2023), and AALasso (Fu, 2023), as well as the overall benchmarking pipeline and visualization framework, were taken from and adapted from the public repository accompanying that study. We extended this framework by incorporating two additional five-nearest-neighbor baselines and by evaluating two per-protein VEFill models, enabling a comprehensive and head-to-head comparison across a broad set of models.

We benchmarked both per-protein VEFill-max and VEFill-min against the following methods: a Residue-mean baseline; Wu et al (2019); Envision; FactorizeDMS; AALasso; and two five-nearest-neighbor approaches that impute variant effects using similarity between amino-acid substitutions at the same residue position. 5NN-BLOSUM100 imputes the effect of a target substitution by identifying the five most similar amino-acid substitutions at the same position, where similarity is quantified using the BLOSUM100 substitution matrix. The imputed score is computed as the median of the experimentally measured DMS scores of these five nearest neighbors, following the approach described in the preprint (Gebbia et al, 2024b). 5NN-Functional is an alternative nearest-neighbor baseline based on an empirically derived functional similarity metric. For each amino-acid substitution type ($X \rightarrow Y$), we computed the mean normalized DMS score of that substitution across all positions within the same protein, thereby capturing its average functional effect in that protein context. For a given target position, the five substitutions with the most similar functional profiles were selected as nearest neighbors, and the imputed score was computed as the mean of their experimental DMS scores. This variant of the 5NN approach was suggested by the original developers of the 5NN-BLOSUM100 method but has not been previously described in the literature.

To ensure fair comparison among models with differing feature requirements, benchmarking was performed under three complementary evaluation settings. First, all 146 proteins were evaluated using all features available to each respective method. Second, a subset of 40 proteins was analyzed for which all benchmarked models could be applied to identical feature sets, enabling a direct, like-for-like comparison. Third, a subset of 68 proteins was used to enable additional cross-model comparisons; however, VEFill-max

could not be applied to this 68-protein set and was therefore compared using the reduced 40-protein subset due to the unavailability of the required EVE scores for some proteins.

To evaluate how imputation performance depends on dataset completeness, we performed a controlled subsampling analysis on 28 high-quality DMS datasets for which all benchmarked models had complete feature availability and overall completeness of at least 94.7%. For each dataset, 10% of variants were held out as a fixed test set, while the remaining variants were subsampled at increasing fractions (10–90% of the full dataset) to simulate partially complete experiments. VEFill dynamically recomputed positional mean DMS scores using only the available training subset at each completeness level, ensuring that no information from held-out variants was used during imputation. Median Pearson correlation across the 28 datasets was used as the summary statistic for all models.

To further assess VEFill's utility in low-data regimes relevant to sparse library design, we evaluated how model performance depends on the number of experimentally measured substitutions available per position. Using the same 28 high-quality DMS datasets employed in the completeness analysis, we trained per-protein VEFill-min and VEFill-max while restricting the training data to a fixed number of mutations per position ($N = 1$–$20$), randomly sampled for each site. Mean DMS scores were recomputed dynamically using only the available mutations at each position, ensuring that no information from held-out variants was used.

## Data availability

The datasets and computer code produced in this study are available in the following databases: Source code: GitHub, VEFill (https://github.com/PlushZ/VEFill). Source code (frozen release used in this study): Zenodo, PlushZ/VEFill: v1.0.1 (Polunina, 2025a), (https://doi.org/10.5281/zenodo.17978457). Processed datasets, database backup, pretrained models, supplementary materials: Zenodo, VEFill (Polunina, 2025b), (https://doi.org/10.5281/zenodo.18223107).

The source data of this paper are collected in the following database record: biostudies:S-SCDT-10_1038-S44320-026-00203-y.

## Peer review information

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

## Acknowledgements

PVP and WM were supported by the Freiburg Galaxy Team funded by the German Federal Ministry of Education and Research BMBF grant 031 A538A de.NBI-RBC and the Ministry of Science, Research and the Arts Baden-Württemberg (MWK) within the framework of LIBIS/de.NBI Freiburg. AFR received funding from NIH/NHGRI grants UM1HG011969 and RM1HG010461. This project received grant funding from the Australian government. The computing infrastructure was partly provided by the de.NBI Cloud within the German Network for Bioinformatics Infrastructure (de.NBI) and ELIXIR-DE (Forschungszentrum Jülich and W-de.NBI-001, W-de.NBI-004, W-de.NBI-008, W-de.NBI-010, W-de.NBI-013, W-de.NBI-014, W-de.NBI-016, W-de.NBI-022).

## Author contributions

**Polina V Polunina**: Conceptualization; Data curation; Software; Formal analysis; Validation; Investigation; Visualization; Methodology; Writing—original draft; Writing—review and editing. **Wolfgang Maier**: Conceptualization; Resources; Supervision; Funding acquisition; Validation; Methodology; Writing—original draft; Project administration; Writing—review and editing. **Alan F Rubin**: Conceptualization; Supervision; Funding acquisition; Validation; Methodology; Writing—original draft; Project administration; Writing—review and editing.

Source data underlying figure panels in this paper may have individual authorship assigned. Where available, figure panel/source data authorship is listed in the following database record: biostudies:S-SCDT-10_1038-S44320-026-00203-y.

## Disclosure and competing interests statement

The authors declare no competing interests.

# Expanded View Figures

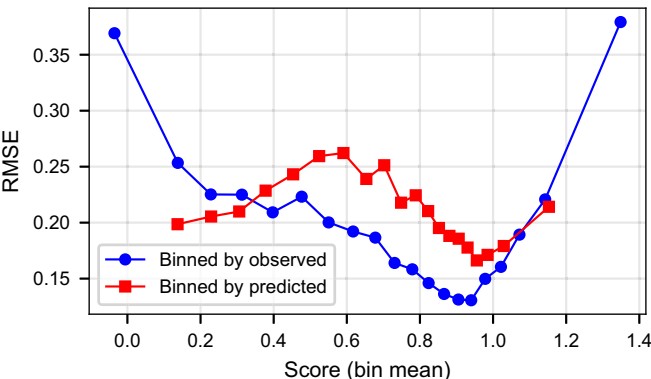

**Figure EV1. RMSE computed in quantile-based bins using either observed or predicted DMS scores.**

The observed-binned curve reveals variation in predictive accuracy across the biological effect spectrum, while the predicted-binned curve reflects model calibration across its output range Source data are available online for this figure.

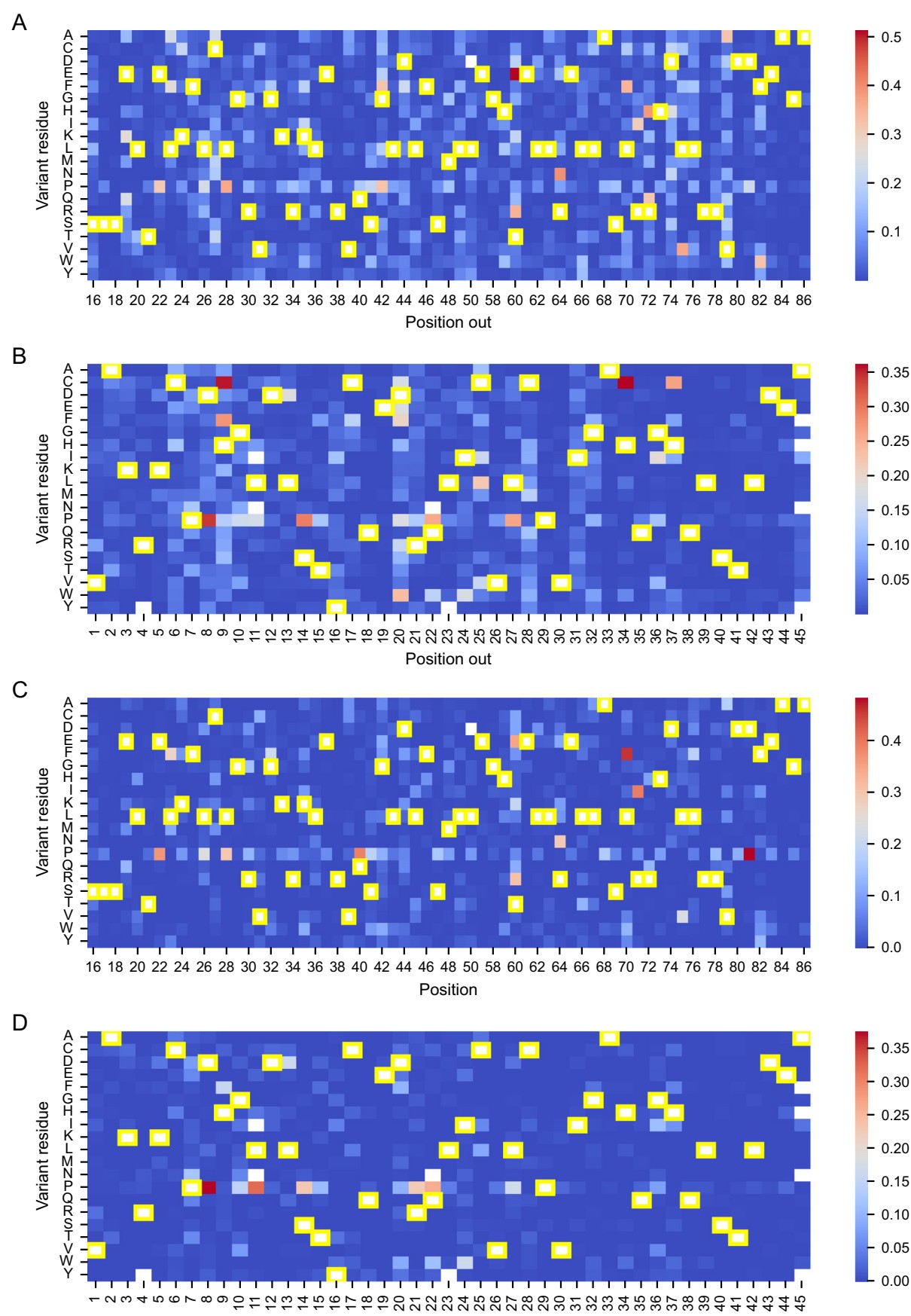

◀ **Figure EV2.   The squared error between predicted and observed normalized DMS scores for the two best-performing per-protein models under LOPosO and LOVarO strategies.**

(**A**, **B**) SE distributions for FADD and TRIM44, respectively, under the LOPosO strategy, where in each iteration, all variants at a single position are held out for testing. The squared error for each variant at the held-out position is calculated and plotted in the heatmap. This process is repeated across all positions, holding out a different position in each iteration. (**C**, **D**) SE distributions for FADD and TRIM44, respectively, using the LOVarO strategy, in which only one variant is held out at a time. The SE is calculated for that single held-out variant and plotted accordingly. This fine-grained approach evaluates the model's ability to predict individual unseen mutations. Cells with yellow frames indicate the wild-type amino acid at each position Source data are available online for this figure.

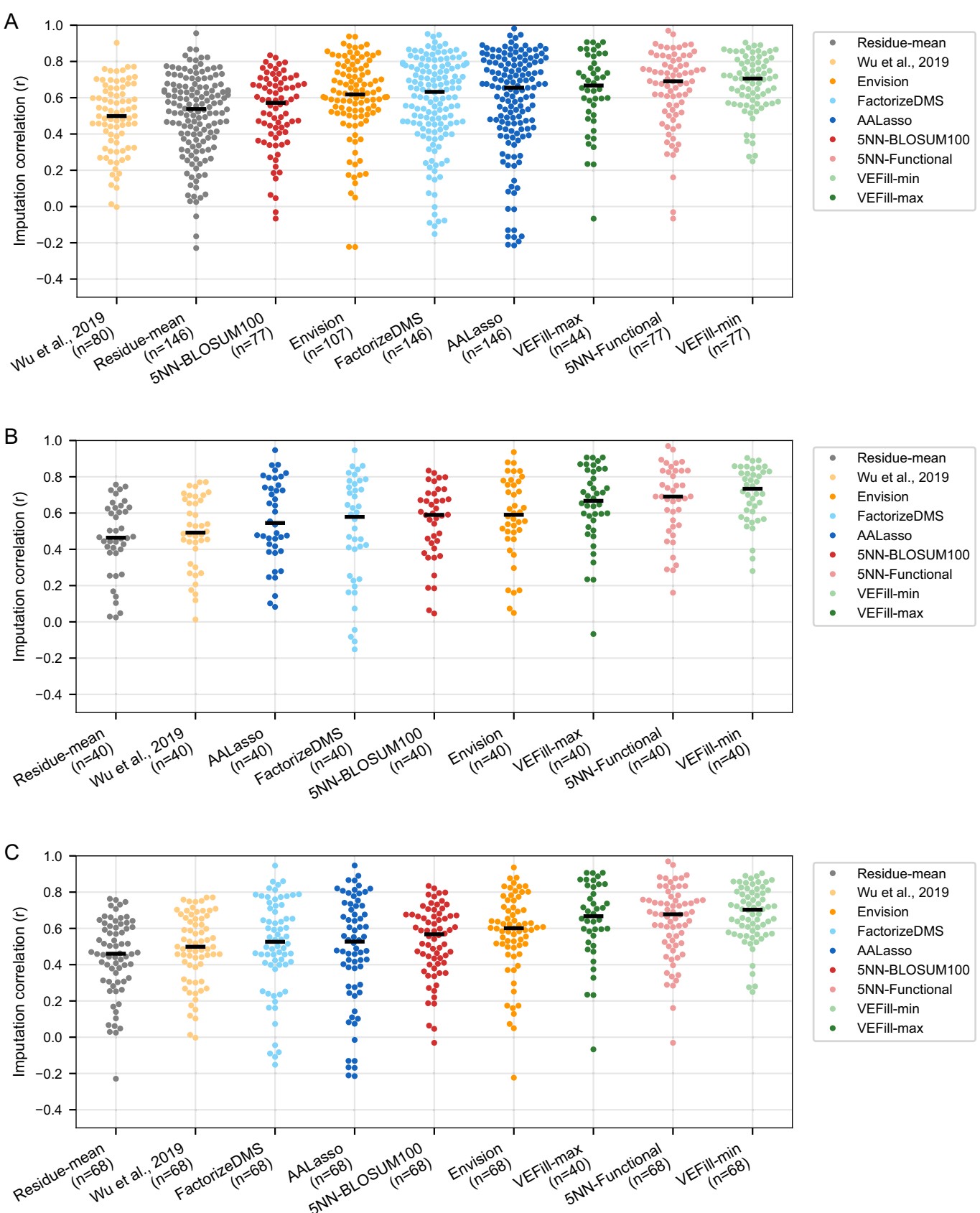

◄  **Figure EV3.  Benchmarking of imputation models across DMS datasets.**

(**A**) All 146 proteins with available features were included in the analysis. Because some models require specific annotations (e.g., structural or evolutionary features), not all models could be trained on all proteins; each method is therefore evaluated on the maximal subset of datasets for which its requisite features were available. (**B**) This analysis includes only the 40 proteins for which all benchmarking models had complete feature availability. Each model was trained and evaluated on the exact same set of proteins, enabling strict, like-for-like comparison without confounding differences in dataset coverage. (**C**) This panel includes the 68 proteins for which all benchmarking models had complete feature coverage. Because VEFill-max requires EVE scores, which were unavailable for a subset of proteins, it could only be trained and evaluated on the 40 proteins (as shown in (**B**)), whereas all other models were evaluated on the full 68-protein set. A simple Residue-mean baseline—imputing a variant's effect as the mean DMS score of all other variants at the same position—is included for reference. Each point in the swarm plot represents the per-protein Pearson correlation between predicted and experimental DMS scores for that model. Differences in point counts across models reflect differences in feature availability across datasets Source data are available online for this figure.

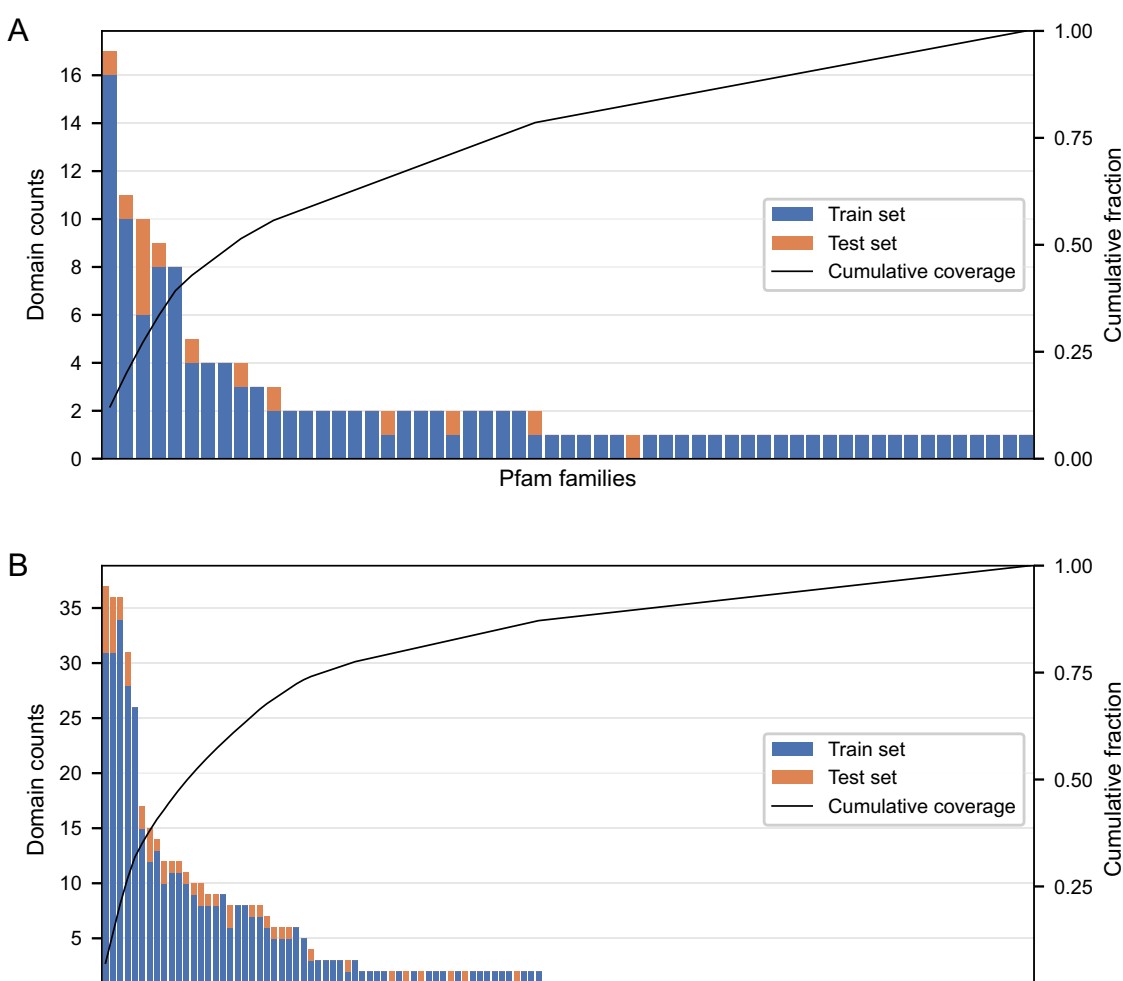

**Figure EV4.   Distribution of protein families in the 140-domain and 521-domain training datasets used for reduced-feature model training.**

(A) 140-domain dataset; (B) 521-domain dataset. The left Y-axis indicates the number of protein domains per protein family. Bars are stacked to show counts in the training and test sets. Pfam IDs are ordered by descending total domain count. The right Y-axis shows the cumulative fraction of the dataset accounted for by each Pfam family, highlighting the dominance of a small subset of Pfam families. (See Dataset EV1 for detailed list of domains, including UniProt IDs, Pfam IDs, and their presence in the 140- and 521-domain datasets) Source data are available online for this figure.

