## [Peer Review File · Molecular Systems Biology]

VEFill: accurate and generalizable deep mutational scanning score imputation across protein domains

Polina Polunina, Wolfgang Maier, and Alan Rubin

Corresponding author(s): Alan Rubin (alan.rubin@unimelb.edu.au)

Review Timeline:

Submission Date:	3rd Sep 25
Editorial Decision:	25th Oct 25
Revision Received:	13th Jan 26
Editorial Decision:	17th Feb 26
Revision Received:	24th Feb 26
Accepted:	6th Mar 26

Editor: Poonam Bheda

Transaction Report:

25th Oct 2025

Manuscript Number: MSB-2025-13319

Title: VEFill: accurate and generalizable deep mutational scanning score imputation across protein domains

Dear Dr Rubin,

Thank you again for submitting your work to Molecular Systems Biology. We have now heard back from the two reviewers who agreed to evaluate your study. As you will see below, the reviewers find that the VEFill method is of potential interest. However, it is difficult to assess the potential utility and impact of the method without appropriate benchmarking and advantages/limitations discussed compared to other methods as mentioned by Reviewer 2. This would be required for a successful revision at MSB. All other issues raised would need to be satisfactorily addressed. Please let me know in case you would like to discuss in further detail any of the comments, I would be happy to schedule a call.

We require:

1) A .docx formatted version of the manuscript text (including legends for main figures, EV figures and tables). Please make sure that the changes are highlighted to be clearly visible. Alternatively you may choose to submit your manuscript as a LaTeX file.

4) A .docx formatted letter INCLUDING the reviewers' reports and your detailed point-by-point responses to their comments. As part of the EMBO Press transparent editorial process, the point-by-point response is part of the Peer Review File (PRF), which will be published alongside your paper.

5) A complete author checklist, which you can download from our author guidelines (<https://www.embopress.org/page/journal/17574684/authorguide#submissionofrevisions>). Please insert information in the checklist that is also reflected in the manuscript. The completed author checklist will also be part of the PRF.

6) Please note that all corresponding authors are required to supply an ORCID ID for their name upon submission of a revised manuscript.

7) It is mandatory to include a 'Data Availability' section after the Materials and Methods. Before submitting your revision, primary datasets produced in this study need to be deposited in an appropriate public database, and the accession numbers and database listed under 'Data Availability'. Please remember to provide a reviewer password if the datasets are not yet public (see <https://www.embopress.org/page/journal/17574684/authorguide#dataavailability>).

In case you have no data that requires deposition in a public database, please state so in this section as follows: "This study includes no data deposited in external repositories". Note that the Data Availability Section is restricted to new primary data that are part of this study.

8) All Materials and Methods need to be described in the main text using our 'Structured Methods' format, which is required for all research articles. According to this format, the Methods section includes a Reagents and Tools Table (listing key reagents, experimental models, software and relevant equipment and including their sources and relevant identifiers) followed by a Methods and Protocols section describing the methods using a step-by-step protocol format. The aim is to facilitate adoption of the methodologies across labs. Please upload the Reagents and Tools table as a separate document when submitting your revised manuscript. More information on how to adhere to this format as well as a downloadable template (.docx) for the Reagents and Tools Table can be found in our author guidelines:

<https://www.embopress.org/page/journal/17444292/authorguide#structuredmethods>

9) For data quantification: please specify the name of the statistical test used to generate error bars and p-values, the number (n) of independent experiments (specify technical or biological replicates) underlying each data point and the test used to calculate p-values in each figure legend. The figure legends should contain a basic description of n, p-values and the test applied. Graphs must include a description of the bars and the error bars (s.d., s.e.m.). Please provide exact p-values (in either the figure or figure legend).

10) Our journal encourages inclusion of *data citations in the reference list* to directly cite datasets that were re-used and obtained from public databases. Data citations in the article text are distinct from normal bibliographical citations and should directly link to the database records from which the data can be accessed. In the main text, data citations are formatted as follows: "Data ref: Smith et al, 2001" or "Data ref: NCBI Sequence Read Archive PRJNA342805, 2017". In the Reference list, data citations must be labeled with "[DATASET]". A data reference must provide the database name, accession number/identifiers and a resolvable link to the landing page from which the data can be accessed at the end of the reference. Further instructions are available at .

11) We replaced Supplementary Information with Expanded View (EV) Figures and Tables that are collapsible/expandable online. EV Figures should be cited as 'Figure EV1, Figure EV2' etc... in the text and their respective legends should be included in the main text after the legends of regular figures.

- Additional Tables/Datasets should be labeled and referred to as Table EV1, Dataset EV1, etc. Legends should be provided in a separate tab in case of .xls files. Alternatively, the legend can be supplied as a separate text file (README) and zipped together with the Table/Dataset file.

<https://www.embopress.org/page/journal/17574684/authorguide#expandedview>

12) Author contributions: CRedit has replaced the traditional author contributions section because it offers a systematic machine-readable author contributions format that allows for more effective research assessment. Please remove the Authors Contributions from the manuscript and use the free text boxes beneath each contributing author's name in our system to add specific details on the author's contribution. More information is available in our guide to authors.

13) Disclosure statement and competing interests: We updated our journal's competing interests policy in January 2022 and request authors to consider both actual and perceived competing interests. Please review the policy <https://www.embopress.org/competing-interests> and update your competing interests if necessary.

14) Every published paper now includes a 'Synopsis' to further enhance discoverability. Synopses are displayed on the journal webpage and are freely accessible to all readers. They include a short stand first (maximum of 300 characters, including space) as well as 2-5 one-sentences bullet points that summarizes the paper. Please write the bullet points to summarize the key NEW findings. They should be designed to be complementary to the abstract - i.e. not repeat the same text. We encourage inclusion of key acronyms and quantitative information (maximum of 30 words / bullet point). Please use the passive voice. Please attach these in a separate file or send them by email, we will incorporate them accordingly.

Please note that these would be the final versions and changes during proofing are usually not allowed.

15) As part of the EMBO Publications transparent editorial process initiative (see our policy here:

https://www.embopress.org/transparent-process#Review_Process), Molecular Systems Biology will publish online a Peer Review File (PRF) to accompany accepted manuscripts.

In the event of acceptance, this file will be published in conjunction with your paper and will include the anonymous referee reports, your point-by-point response and all pertinent correspondence relating to the manuscript. Let us know whether you agree with the publication of the PRF and as here, if you want to remove or not any figures from it prior to publication.

Please note that the Author checklist will be published at the end of the PRF.

Molecular Systems Biology has a "scooping protection" policy, whereby similar findings that are published by others during review or revision are not a criterion for rejection. Should you decide to submit a revised version, I do ask that you get in touch after three months if you have not completed it, to update us on the status.

Yours sincerely,

Poonam Bheda, PhD
Scientific Editor
Molecular Systems Biology

Reviewer #1:

In this work, Polunina, Maier, and Rubin develop VEFill, an approach to impute variant effect scores based on complementary data from protein language model embeddings, sequence statistics, and physicochemical properties. Their model is trained on a large DMS data set called Human Domainome 1, which includes data from 522 protein domains (reduced, in some cases, to a smaller set of 140 domains for more detailed models that incorporate scores computed with the EVE model). The authors used a gradient boosting decision tree as their model for imputation.

Overall, this paper makes an interesting contribution to the literature on predicting and interpreting variant effects. The authors consider multiple different training conditions, assessing how different features contribute to predictive power. Their imputation model, VEFill, also performs quite well overall, which could make it a useful tool especially for those interested in estimating mutational effects on protein stability.

There are, however, a number of practical and technical questions that should be addressed in the current version of the paper.

Main comments

1. In the main data set (Fig. 2), the authors found that there are two main factors that improve the predictive power of the model: mean DMS scores per position and ESM-1v embeddings. These two sets of features were at least partially complementary, such that adding both together yielded a better model than either one alone. Additional features provided only very small improvements in the model. However, when looking at the top 10 proteins for which the model appears to work best (Figs. 4, 5), it seems that ESM-1v embeddings can be withheld from the model without harming performance. Why is this the case? Is it that the mean DMS scores alone are highly predictive, so that there is little to be gained with additional predictors? Or are the other predictors (EVE scores, ...) able to collectively account for the predictive power of ESM-1v embeddings for these proteins? To explore this last point, it would be interesting to see a version of Fig. 2 where mean DMS scores were combined with the other predictors except for ESM-1v embeddings.

More generally, this difference in typical performance in Fig. 2 vs. Fig. 5 suggests that the relative predictive power of different features isn't the same for every protein. Are there additional factors that could explain when certain features will be more or less useful for predicting variant effects? For example, one might imagine that variant effects for protein families that are more highly represented in training data sets for sequence models - including language models like ESM-1v or other types like EVE - might be easier to predict using sequence features.

2. It is interesting that a roughly 4-fold increase in the size of the training data (140 -> 521 domains for the ESM-1v + mean DMS model) did not typically yield a significant increase in performance for the model. The authors provide some explanation for this, citing multiple inclusions of the same Pfam family within the data set, but they should expand on this point. If one attempts to remove "duplicate" domains, is there a significant increase in performance when going from the smaller (140 proteins if duplicates are included) to the larger data set? In general, it seems like the model is able to make good predictions without a large amount of data. In Fig. 5A, the consistent drop in performance from 85%/15% train/test data to 15%/85% is noted, but in many cases this drop appears rather slight.

3. Related to the question above, can the authors define an accuracy ceiling for this problem? Mutation effect estimates in DMS studies can be noisy, with significant variation even between experimental replicates. Given the inherent noise in experimental data, how accurate could one expect their imputations to be?

4. I understand the use case for this method if one is studying proteins for which DMS studies have not been performed. However, the application when partial DMS scores are available is less clear. Could the authors identify some example cases of DMS/MAVE data sets for which mean DMS scores would be available to use for imputation, but where there is also a significant amount of missing data (and thus, a strong motivation to use imputation to estimate missing values)?

Minor comment: In some places, text in the figures is small and difficult to read. This is most apparent in Figs. 5, 6, and 9, but it affects other figures as well. It would be helpful for the authors to revise these figures to improve readability.

Reviewer #2:

In their manuscript, Polunina et al introduce VEFill, a new machine learning model for DMS score imputation. Imputation is a recurrent need in DMS assays, where it is not rare for some variants to drop out during library cloning or sequencing. The authors train their model on the human domainome dataset, where it performs well even when using a limited set of model

parameters. As I am not an expert in machine learning algorithms, my comments focus on the usefulness of their approach in the context of DMS assays and the characteristics of the data they produce. Regarding the goals the authors set out in the abstract and introduction, here are my main comments:

1. The authors mention several existing models for imputation already exist, but do not provide any comparison in terms of performance between VEFill. Different models can have different strengths and weaknesses: readers should be able to evaluate how VEFill compares to other published models in different scenarios. The lack of a direct comparison is not explained or justified. This is not to say that VEFill would need to be superior to other models in order to be relevant, just that it should at least be compared.

2. The authors do not demonstrate the usefulness of VEFill for designing sparse variant libraries in "low-data regimes". However, as discussed in the manuscript and shown by the 0-shot prediction performances, the mean DMS score for a position is the most important feature in the model. Obtaining the mean DMS score requires performing a full DMS experiment, not one using a sparse library. And so to get to the point where one would be able to design a sparse library, one has to build and screen a full library. Is there a minimal number of position-wise score measurements required for the mean DMS score to be informative? Do data from certain substitutions provide more information? This would be informative to determine if the model can help design sparse libraries.

Other comments:

3. Figure 3: Why are predicted values on average higher? Please scale the x and y axes so they are identical, indicate regression slope, origin, and P-value, and provide a reference to what a 1 to 1 relationship would look like. This applies to the other scatter plots showing predicted vs measured scores.

4. Also Figure 3: Because of the high density of correctly predicted neutral variants, it's possible that the mean RMSE varies substantially along the range of predicted scores. Could the authors provide a binned or rolling RMSE across the full range of predictions?

5. Also Figure 3: Then, based on these RMSE distributions, are the authors confident that VEFill can provide reliable score imputation beyond determining if a variant is benign?

6. (Finally) Also Figure 3: DMS assays can be quite noisy. Does the error in predictions scale with DMS score measurement error? If so, would the model benefit from excluding low confidence scores? Alternatively, could the model also be used to flag positions where a DMS score with high experimental error has a high chance of being erroneous?

7. Figure 4b: Are the best predicted domains shown here from overrepresented domain families in the human domainome dataset (as mentioned later in the text)?

8. Table 1: The model shows high variability in terms of performance, even within assay categories. With the high number of datasets available in MaveDB, is there any reason a more systematic evaluation was not performed? This might allow for a better understanding of scenarios where the model is expected to perform well or not. This is especially true of the DMSscore+ESMv1 embeddings model, which requires very little data on the protein sequence of interest.

9. Also Table 1: For some proteins in MAVEdb (like PTEN), both variant abundance and activity data are available for the same DMS library. Have the authors compared model performance on both?

10. To improve manuscript clarity, I would suggest reducing the number of main text figures and providing data visualization with more depth. For example, Figure 6 presents 4 different heatmaps, which could be condensed into two scatter plots, with labels showing discrepancies between LOPosO and LOVarO.

25th Oct 2025

Manuscript Number: MSB-2025-13319

Title: VEFill: accurate and generalizable deep mutational scanning score imputation across protein domains

Author: Polina V Polunina

Wolfgang Maier

Alan F Rubin

Dear Dr Rubin,

Thank you again for submitting your work to Molecular Systems Biology. We have now heard back from the two reviewers who agreed to evaluate your study. As you will see below, the reviewers find that the VEFill method is of potential interest. However, it is difficult to assess the potential utility and impact of the method without appropriate benchmarking and advantages/limitations discussed compared to other methods as mentioned by Reviewer 2. This would be required for a successful revision at MSB. All other issues raised would need to be satisfactorily addressed. Please let me know in case you would like to discuss in further detail any of the comments, I would be happy to schedule a call.

We require:

1) A .docx formatted version of the manuscript text (including legends for main figures, EV figures and tables). Please make sure that the changes are highlighted to be clearly visible. Alternatively you may choose to submit your manuscript as a LaTeX file.

We have formatted our revised manuscript text as a LaTeX file. Updated changes are highlighted in yellow using \hl. We also moved the Methods to follow the Discussion in line with journal submission guidelines (previously the Methods were after the Introduction).

2) Individual production quality figure files as .eps, .tif, .jpg (one file per figure). For guidance, download the 'Figure Guide PDF'

(<https://www.embopress.org/page/journal/17574684/authorguide#figureformat>).

We have provided vector versions of all figures using high-quality PDF format as described in the Figure Guide, as we ran into trouble with font substitutions when saving in EPS format. Figure fonts have been updated to use the fonts specified in the PDF (or as close as possible).

Thank you. We have prepared source data files and are ready for additional guidance.

4) A .docx formatted letter INCLUDING the reviewers' reports and your detailed point-by-point responses to their comments. As part of the EMBO Press transparent editorial process, the point-by-point response is part of the Peer Review File (PRF), which will be published alongside your paper.

This content is below.

5) A complete author checklist, which you can download from our author guidelines (<https://www.embopress.org/page/journal/17574684/authorguide#submissionofrevisions>). Please insert information in the checklist that is also reflected in the manuscript. The completed author checklist will also be part of the PRF.

We have provided the completed author checklist.

6) Please note that all corresponding authors are required to supply an ORCID ID for their name upon submission of a revised manuscript.

The corresponding author's ORCID ID is included in the portal.

7) It is mandatory to include a 'Data Availability' section after the Materials and Methods. Before submitting your revision, primary datasets produced in this study need to be deposited in an appropriate public database, and the accession numbers and database listed under 'Data Availability'. Please remember to provide a reviewer password if the datasets are not yet public (see

<https://www.embopress.org/page/journal/17574684/authorguide#dataavailability>).

In case you have no data that requires deposition in a public database, please state so in this section as follows: "This study includes no data deposited in external repositories". Note that the Data Availability Section is restricted to new primary data that are part of this study.

Datasets are listed under Data Availability and are publicly accessible.

8) All Materials and Methods need to be described in the main text using our 'Structured Methods' format, which is required for all research articles. According to this format, the Methods section includes a Reagents and Tools Table (listing key reagents, experimental models, software and relevant equipment and including their sources and relevant identifiers) followed by a Methods and Protocols section describing the methods using a step-by-step protocol format. The aim is to facilitate adoption of the methodologies across labs. Please upload the Reagents and Tools table as a separate document when submitting your revised manuscript. More information on how to adhere to this format as well as a downloadable template (.docx) for the Reagents and Tools Table can be found in our author guidelines:

<https://www.embopress.org/page/journal/17444292/authorguide#structuredmethods>

We have included a Reagents and Tools Table listing the relevant software packages and associated versions used for the study.

9) For data quantification: please specify the name of the statistical test used to generate error bars and p-values, the number (n) of independent experiments (specify technical or biological replicates) underlying each data point and the test used to calculate p-values in each figure legend. The figure legends should contain a basic description of n, p-values and the test applied. Graphs must include a description of the bars and the error bars (s.d., s.e.m.). Please provide exact p-values (in either the figure or figure legend).

Statistical details are included in the figures and manuscript text as appropriate.

10) Our journal encourages inclusion of *data citations in the reference list* to directly cite datasets that were re-used and obtained from public databases. Data citations in the article text are distinct from normal bibliographical citations and should directly link to the database records from which the data can be accessed. In the main text, data

citations are formatted as follows: "Data ref: Smith et al, 2001" or "Data ref: NCBI Sequence Read Archive PRJNA342805, 2017". In the Reference list, data citations must be labeled with "[DATASET]". A data reference must provide the database name, accession number/identifiers and a resolvable link to the landing page from which the data can be accessed at the end of the reference. Further instructions are available at <https://www.embopress.org/page/journal/17574684/authorguide#referencesformat>.

Dataset citations have been added to Table 1 and Table 2.

Due to the large number of other dataset records from MaveDB that were used in the study, we have included this information (accession number, resolvable link) in the appropriate EV tables for most datasets.

11) We replaced Supplementary Information with Expanded View (EV) Figures and Tables that are collapsible/expandable online. EV Figures should be cited as 'Figure EV1, Figure EV2' etc... in the text and their respective legends should be included in the main text after the legends of regular figures.

- Additional Tables/Datasets should be labeled and referred to as Table EV1, Dataset EV1, etc. Legends should be provided in a separate tab in case of .xls files. Alternatively, the legend can be supplied as a separate text file (README) and zipped together with the Table/Dataset file.

<https://www.embopress.org/page/journal/17574684/authorguide#expandedview>

We have updated the figures and tables to refer to EV and Appendix as appropriate.

12) Author contributions: CRediT has replaced the traditional author contributions section because it offers a systematic machine-readable author contributions format that allows for more effective research assessment. Please remove the Authors Contributions from the manuscript and use the free text boxes beneath each contributing author's name in our system to add specific details on the author's contribution. More information is available in our guide to authors.

CRediT contributor roles have been included for all authors in the submission system.

13) Disclosure statement and competing interests: We updated our journal's competing interests policy in January 2022 and request authors to consider both actual and perceived competing interests. Please review the policy <https://www.embopress.org/competing-interests> and update your competing interests if necessary.

We have reviewed the policy and have no competing interests.

14) Every published paper now includes a 'Synopsis' to further enhance discoverability. Synopses are displayed on the journal webpage and are freely accessible to all readers. They include a short stand first (maximum of 300 characters, including space) as well as 2-5 one-sentences bullet points that summarizes the paper. Please write the bullet points to summarize the key NEW findings. They should be designed to be complementary to the abstract - i.e. not repeat the same text. We encourage inclusion of key acronyms and quantitative information (maximum of 30 words / bullet point). Please use the passive voice. Please attach these in a separate file or send them by email, we will incorporate them accordingly.

Please note that these would be the final versions and changes during proofing are usually not allowed.

We have provided a synopsis adhering to these guidelines as well as a visual abstract.

15) As part of the EMBO Publications transparent editorial process initiative (see our policy here: https://www.embopress.org/transparent-process#Review_Process), Molecular Systems Biology will publish online a Peer Review File (PRF) to accompany accepted manuscripts.

In the event of acceptance, this file will be published in conjunction with your paper and will include the anonymous referee reports, your point-by-point response and all pertinent correspondence relating to the manuscript. Let us know whether you agree with the publication of the PRF and as here, if you want to remove or not any figures from it prior to publication.

Please note that the Author checklist will be published at the end of the PRF.

We acknowledge this and are very happy to have the PRF included.

Molecular Systems Biology has a "scooping protection" policy, whereby similar findings that are published by others during review or revision are not a criterion for rejection. Should you decide to submit a revised version, I do ask that you get in touch after three months if you have not completed it, to update us on the status.

Yours sincerely,

Poonam Bheda, PhD
Scientific Editor
Molecular Systems Biology

Reviewer #1:

In this work, Polunina, Maier, and Rubin develop VEFill, an approach to impute variant effect scores based on complementary data from protein language model embeddings, sequence statistics, and physicochemical properties. Their model is trained on a large DMS data set called Human Domainome 1, which includes data from 522 protein domains (reduced, in some cases, to a smaller set of 140 domains for more detailed models that incorporate scores computed with the EVE model). The authors used a gradient boosting decision tree as their model for imputation.

Overall, this paper makes an interesting contribution to the literature on predicting and interpreting variant effects. The authors consider multiple different training conditions, assessing how different features contribute to predictive power. Their imputation model, VEFill, also performs quite well overall, which could make it a useful tool especially for those interested in estimating mutational effects on protein stability.

There are, however, a number of practical and technical questions that should be addressed in the current version of the paper.

We thank the reviewer for this positive assessment of the work and appreciate the feedback. Our detailed point by point response is below in blue.

Main comments

1. In the main data set (Fig. 2), the authors found that there are two main factors that improve the predictive power of the model: mean DMS scores per position and ESM-1v embeddings. These two sets of features were at least partially complementary, such that adding both together yielded a better model than either one alone. Additional features provided only very small improvements in the model. However, when looking at the top 10 proteins for which the model appears to work best (Figs. 4, 5), it seems that ESM-1v embeddings can be withheld from the model without harming performance. Why is this the case? Is it that the mean DMS scores alone are highly predictive, so that there is little to be gained with additional predictors? Or are the other predictors (EVE scores, ...) able to collectively account for the predictive power of ESM-1v embeddings for these proteins? To explore this last point, it would be interesting to see a version of Fig. 2 where mean DMS scores were combined with the other predictors except for ESM-1v embeddings.

More generally, this difference in typical performance in Fig. 2 vs. Fig. 5 suggests that the relative predictive power of different features isn't the same for every protein. Are there additional factors that could explain when certain features will be more or less useful for predicting variant effects? For example, one might imagine that variant effects for protein families that are more highly represented in training data sets for sequence models - including language models like ESM-1v or other types like EVE - might be easier to predict using sequence features.

We thank the reviewer for this thoughtful comment. We would like to clarify that Fig. 2 and Figs. 4–5 correspond to two different modeling settings. Fig. 2 evaluates a general cross-protein model trained on all 140 proteins, whereas Fig. 5(C) evaluates per-protein models trained separately on each protein with an 80/20 train/test split.

In the cross-protein setting (Fig. 2), the model must generalize to proteins that were not used for training, and the positional mean DMS score is not available during training for unseen proteins. In this regime, ESM-1v embeddings are essential, as they provide cross-protein sequence information that strongly boosts performance: most of the gain beyond MeanDMS alone comes from adding ESM-1v.

By contrast, in the per-protein setting of Fig. 5(C), the model is trained on 80% of the mutations for a single protein. This means that, for most positions, multiple substitutions are already observed in the training set, so the mean DMS score per position becomes extremely informative, and ESM-1v barely adds information on top of this strong positional signal. This explains why withholding ESM-1v in Fig. 5(C) does not substantially harm performance.

Importantly, in low-data regimes - when only a small number of mutations per position are available - the situation is different. In those cases, the positional mean is much less reliable, and ESM-1v can complement MeanDMS, helping to maintain model performance when experimental coverage is sparse.

Following the reviewer’s suggestion, we have added an extra bar in Fig. 2 for the model that combines the mean DMS score with all other non-ESM features: EVE, physicochemical properties, substitution matrices, SNV indicator ($R^2=0.606$) - bar labeled “eve+25sm+pc+csv+mean_dms”.

Figure 2. General model performance (R^2) for DMS score imputation across feature sets. Models were trained on the 140 protein domains for which EVE scores were available. Predictive power increased with feature richness, especially when ESM-1v embeddings and positional priors (mean DMS scores) were included. The top model ($R^2 = 0.64$, $r = 0.80$) excluded only the ESM-1v difference vector (see Table EV2 for full definitions of feature sets, and Appendix Fig. S2 and Table EV3 for full results).

We added text to the “Results” section “General model evaluation” subsection as highlighted below:

We evaluated performance across various biologically motivated feature combinations, including evolutionary scores (EVE), substitution matrices, physicochemical properties, and ESM-1v sequence embeddings. Simpler models using individual feature types—such as only EVE scores, substitution matrices, or biochemical descriptors—exhibited limited performance ($R^2 < 0.15$). Incorporating ESM-1v embeddings led to a substantial improvement ($R^2 = 0.42$), highlighting the utility of learned contextual sequence representations. A model relying solely on mean DMS scores per position, without any additional external features, performed better than these minimal configurations and even outperformed the model using only ESM-1v embeddings, achieving an R^2 of 0.53. However, it still underperformed relative to the feature-set configuration that combined mean DMS with all non-ESM features ($R^2 = 0.61$). The model incorporating both mean DMS and ESM-1v embeddings ($R^2 = 0.63$) achieved better performance than all aforementioned configurations, indicating that learned embeddings and position-specific priors capture complementary information. These results further demonstrate that non-ESM features cannot fully compensate for the contextual information provided by ESM-1v in the cross-protein setting, where models must generalize to protein domains entirely absent from training. In this context, the positional mean DMS score is not available for unseen proteins, making sequence-derived representations essential.

We also added text to the “Results” section “Per-protein model evaluation: random splits, LPosO, split by SNV, split by AA class” subsection as highlighted below:

We also tested how reduced feature sets influenced model performance. Models trained using only ESM-1v and mean DMS scores, as well as models trained without ESM-1v embeddings, retained competitive performance compared to models trained on full feature set (Fig. 5(C)). The limited contribution of ESM-1v in this per-protein context reflects the fact that, under random 80/20 splits, most positions are already represented in the training data. Consequently, the positional mean DMS score acts as a strong empirical prior that captures much of the site-specific mutational tolerance, leaving little residual variance for contextual sequence embeddings to explain. This stands in contrast to the cross-protein setting (Fig. 2), where positional means are unavailable for new proteins and ESM-1v plays a central role in enabling generalization.

Additionally, to avoid confusion between general cross-protein and per-protein models, we now use the term “general cross-protein model” instead of “general model.” For example, we renamed the titles of subsections in the “Methods” and “Results” sections that are dedicated to the general cross-protein setting by explicitly adding “cross-

protein” to the titles. We also use this terminology consistently throughout the manuscript. For the per-protein setting, we retain the term “per-protein model.”

2. It is interesting that a roughly 4-fold increase in the size of the training data (140 → 521 domains for the ESM-1v + mean DMS model) did not typically yield a significant increase in performance for the model. The authors provide some explanation for this, citing multiple inclusions of the same Pfam family within the data set, but they should expand on this point. If one attempts to remove “duplicate” domains, is there a significant increase in performance when going from the smaller (140 proteins if duplicates are included) to the larger data set? In general, it seems like the model is able to make good predictions without a large amount of data. In Fig. 5A, the consistent drop in performance from 85%/15% train/test data to 15%/85% is noted, but in many cases this drop appears rather slight.

We thank the reviewer for this insightful comment. To further clarify whether redundancy arising from multiple domains belonging to the same Pfam family inflates model performance—a concern raised by the reviewer—we extended our original analysis of Pfam-level overlap (see Fig. R1 below). First, we retrained the full-feature model using only one representative protein domain per Pfam family from the 140-domain “reduced Domainome” dataset (57 unique Pfam families). Removing duplicates resulted in only a negligible change in performance ($\Delta\text{RMSE} \approx 0.0026$ for 57 Pfam-unique vs. 140 Pfam-redundant), indicating that redundancy contributes minimally to model accuracy.

We next compared minimal-feature models (ESM+MeanDMS and MeanDMS alone) trained using Pfam-unique train/test splits derived from the 57 Pfam families represented in the 140-domain dataset with models trained using Pfam-unique splits across the 127 Pfam families present in the full 521-domain Human Domainome 1 dataset. Increasing Pfam diversity from 57 to 127 families unexpectedly yielded slightly higher RMSEs (0.2219 → 0.2654 for ESM+MeanDMS and 0.2464 → 0.3671 for MeanDMS), in comparison to the modest changes observed when using the corresponding non-Pfam-unique datasets (140 domains → 521 domains: 0.2166 → 0.2039 and 0.2443 → 0.3012 for ESM+MeanDMS and MeanDMS alone, respectively).

Together, these results suggest that model performance largely plateaus once a moderate level of structural diversity is captured, and that neither dataset size nor protein family redundancy are primary drivers of accuracy. Instead, the dominant factor appears to be the representation of a sufficient number of distinct Pfam families.

Figure R1. Effect of dataset expansion on model performance. Bars show the change in RMSE (Δ RMSE) when increasing the training dataset by either adding additional domains from the same Pfam families (57→140 and 127→521), adding new Pfam-unique families (57→127), or expanding the set of Pfam-redundant domains from 140 total to all 521 domains in Human Domainome 1 (140→521). Changes are shown separately for the full-feature model, the ESM1v+MeanDMS model, and the MeanDMS-only model.

3. Related to the question above, can the authors define an accuracy ceiling for this problem? Mutation effect estimates in DMS studies can be noisy, with significant variation even between experimental replicates. Given the inherent noise in experimental data, how accurate could one expect their imputations to be?

We thank the reviewer for this important question. In response, we have added a new subsection “Estimating the accuracy ceiling imposed by DMS assay noise” in section “Results” to the main manuscript that formally estimates the accuracy ceiling imposed by DMS assay noise, evaluates VEFill’s performance relative to this limit, and examines whether excluding high-uncertainty measurements improves prediction accuracy. This analysis shows that VEFill performs close to the noise-imposed ceiling, and filtering low-confidence variants does not provide a measurable benefit. The newly added subsection to the “Methods” is below:

Estimation of the DMS assay noise ceiling

Deep mutational scanning measurements contain inherent experimental variability, which imposes an upper bound on the predictive accuracy achievable by any computational model. To quantify this limit, we estimated a noise ceiling for each DMS dataset based on the assay-reported measurement uncertainty (e.g., per-variant standard errors or confidence estimates). The objective of this analysis is to determine the maximum correlation that could be expected between two repeated measurements of the same experiment and thereby define the upper limit against which VEFill's performance should be interpreted.

Let s_i denote the reported normalized DMS score for variant i , and let σ_i denote its corresponding measurement uncertainty. For each protein domain, we simulated $B = 300$ replicate experiments by adding heteroscedastic Gaussian noise to each variant according to its reported uncertainty:

$\epsilon_i^{(b)}$ is drawn from a normal distribution with mean 0 and variance σ_i^2 ,
 $s_i^{(b)} = s_i + \epsilon_i^{(b)}$, for $b = 1, \dots, B$

For each simulated replicate dataset, we computed the Pearson correlation between the original measurements and the noisy replicate:

$$r^{(b)} = \text{corr}(s, s^{(b)})$$

The distribution $\{ r^{(b)} \}$ for $b = 1$ to B reflects the expected agreement between two independent realizations of the same experimental assay under its measured noise characteristics. We define the noise ceiling for a given protein domain as the mean of this distribution.

This procedure was applied to 139 of the 140 feature-complete Human Domainome 1 protein domains. One domain (LIMS2 zinc finger, LIM-type domain) contained too few variants in its test set under the VEFill 80/20 split to support a stable Pearson correlation estimate and was therefore excluded from this analysis.

The newly added subsection to the "Results" is below:

Performance evaluation relative to the DMS assay noise ceiling

Because experimental noise imposes a fundamental upper bound on achievable predictive accuracy, we used the noise ceiling estimation procedure to evaluate per-protein VEFill's performance relative to the maximum accuracy permitted by DMS assay variability. Across the 139 domains (after excluding one domain with

too few variants), noise ceilings ranged from 0.68 to 0.99, reflecting differences in experimental precision across assays. VEFill's per-protein Pearson correlations ($r = 0.52\text{--}0.91$ under the 80/20 split) lie consistently below, but generally close to, these ceilings (Appendix Fig. S4). Occasional cases in which VEFill slightly exceeds the estimated ceiling are expected due to sampling variability in both the simulated replicates and the correlation estimates. Collectively, these results demonstrate that VEFill performs near the accuracy limit imposed by experimental noise, and that much of the remaining discrepancy between predicted and observed scores is attributable to measurement uncertainty rather than model underfitting.

Although measurement uncertainty imposes an upper limit on achievable accuracy, we additionally evaluated whether filtering out variants with the highest reported assay uncertainty improves model performance. For each protein domain, we progressively removed the top 10%, 20%, and 30% most uncertain variants and retrained the per-protein models. Median Pearson correlations remained effectively unchanged across all filtering levels (Appendix Fig. S5), indicating that VEFill does not substantially benefit from excluding low-confidence measurements. Nonetheless, large discrepancies between VEFill predictions and experimentally reported scores—particularly for variants with high assay uncertainty—may help flag measurements that warrant further experimental verification.

Additionally, we added text to the “Discussion”:

Across multiple benchmarking analyses, per-protein VEFill demonstrates robust and competitive performance relative to a broad range of existing DMS imputation methods, including residue-wise mean baselines, Wu et al. (2019), Envision, FactorizeDMS, AALasso, and two five-nearest-neighbor approaches. In controlled completeness-based benchmarking, two different VEFill settings consistently outperform all previously published models once at least 20% of variants have been experimentally measured. Importantly, noise-ceiling analyses indicate that per-protein VEFill operates near the accuracy limit imposed by experimental variability in DMS assays, suggesting that further performance gains are primarily constrained by measurement uncertainty rather than model capacity.

4. I understand the use case for this method if one is studying proteins for which DMS studies have not been performed. However, the application when partial DMS scores are available is less clear. Could the authors identify some example cases of DMS/MAVE data sets for which mean DMS scores would be available to use for

imputation, but where there is also a significant amount of missing data (and thus, a strong motivation to use imputation to estimate missing values)?

We thank the reviewer for this helpful question. To clarify the practical use of VEFill in settings where partial DMS measurements are available, we have added a new subsection to the main manuscript entitled “Completeness of existing DMS datasets and practical motivation for VEFill.” In this subsection, we analyze completeness across 841 contemporary MaveDB score sets and identify representative datasets in which positional mean DMS scores can be computed despite substantial fractions of substitutions remaining unmeasured. We further show that many such datasets fall within the completeness regime ($\geq 20\%$) where our benchmarking demonstrates that VEFill provides the strongest benefit. The newly added manuscript text is reproduced below:

Completeness of existing DMS datasets and practical motivation for VEFill

To evaluate the practical relevance of VEFill in settings where partial DMS measurements are available, we conducted a completeness analysis inspired by the framework of Fu et al. (2023). Many contemporary DMS experiments report only a subset of all possible single–amino-acid substitutions, resulting in datasets where positional mean DMS scores can be computed but substantial fractions of variants remain unmeasured. Such settings align closely with VEFill's intended use case—augmenting partially complete experimental maps with accurate imputations for the remaining substitutions.

We analyzed 841 public score sets from the most recent MaveDB Zenodo release (MaveDB contributors, 2025). For each dataset, we quantified completeness using a modified version of Fu's formulation, defining the total number of possible substitutions at each scanned residue as 20 (the 20 standard amino acids, including the termination substitution, minus the wild-type residue). Completeness was calculated as the number of reported single–amino-acid variants divided by the number of scanned residues multiplied by 20.

Each dataset was then assigned to one of eleven completeness bins spanning the interval from 0 to 1 in increments of 0.1. This analysis revealed that partially complete datasets are very common: many experiments report 20–70% of possible substitutions, while only a minority achieve near-complete coverage (Table EV4).

Importantly, our benchmarking results (Fig. 7; Fig. EV3) demonstrate that VEFill begins to outperform alternative imputation strategies once at least 20% of variants have been experimentally measured, corresponding to a completeness

of 0.2 or greater. When combined with the completeness distribution observed across MaveDB, this indicates that a large proportion of real-world DMS datasets fall squarely within the regime where VEFill provides the most substantial benefit (Table 1).

Minor comment: In some places, text in the figures is small and difficult to read. This is most apparent in Figs. 5, 6, and 9, but it affects other figures as well. It would be helpful for the authors to revise these figures to improve readability.

We thank the reviewer for this helpful suggestion. We have revised multiple figures to improve readability, including Figs. 5, 6, and 9, by increasing font sizes, adjusting layout and spacing, and improving visual clarity where needed.

Reviewer #2:

In their manuscript, Polunina et al introduce VEFill, a new machine learning model for DMS score imputation. Imputation is a recurrent need in DMS assays, where it is not rare for some variants to drop out during library cloning or sequencing. The authors train their model on the human domainome dataset, where it performs well even when using a limited set of model parameters. As I am not an expert in machine learning algorithms, my comments focus on the usefulness of their approach in the context of DMS assays and the characteristics of the data they produce. Regarding the goals the authors set out in the abstract and introduction, here are my main comments:

1. The authors mention several existing models for imputation already exist, but do not provide any comparison in terms of performance between VEFill. Different models can have different strengths and weaknesses: readers should be able to evaluate how VEFill compares to other published models in different scenarios. The lack of a direct comparison is not explained or justified. This is not to say that VEFill would need to be superior to other models in order to be relevant, just that it should at least be compared.

We thank the reviewer for this thoughtful comment. In the revised manuscript, we now include a comprehensive benchmarking analysis comparing VEFill against a broad range of existing imputation methods. These comparisons are summarized in Fig. 7, Fig. EV3 and described in the “Methods” subsection “Benchmarking framework and comparative evaluation setup” and in the “Results” subsection “Benchmarking against existing DMS imputation methods” subsection. The newly added subsection to the “Methods” is provided below:

Benchmarking framework and comparative evaluation setup

To enable a direct and reproducible comparison with existing DMS imputation approaches, we adopted and extended the benchmarking framework developed by Fu et al. (2023). Specifically, the selection of the 146 DMS datasets and the 28 high-quality datasets used for benchmarking, the implementations of the residue-mean baseline, Wu et al. (2019), Envision (Gray et al., 2018), FactorizeDMS (Fu et al., 2023), and AALasso (Fu et al., 2023), as well as the overall benchmarking pipeline and visualization framework, were adapted from the public repository accompanying that study. We extended this framework by incorporating two additional five-nearest-neighbor baselines and by evaluating two per-protein VEFill models, enabling a comprehensive and head-to-head comparison across a broad set of models.

We benchmarked both per-protein VEFill-max (the full-feature model) and VEFill-min (a reduced-feature model using only the positional mean DMS score and ESM-1v embeddings for the wild-type, variant, and their difference) against the following methods: a residue-mean baseline; Wu et al. (2019); Envision; FactorizeDMS; AALasso; and two five-nearest-neighbor approaches that impute variant effects using similarity between amino-acid substitutions at the same residue position. The 5NN-BLOSUM100 method imputes the effect of a target substitution by identifying the five most similar amino-acid substitutions at the same position, where similarity is quantified using the BLOSUM100 substitution matrix. The imputed score is computed as the median of the experimentally measured DMS scores of these five nearest neighbors, following the approach described by Gebbia et al. (2024). The 5NN-Functional method is an alternative nearest-neighbor baseline based on an empirically derived functional similarity metric. For each amino-acid substitution type ($X \rightarrow Y$), we computed the mean normalized DMS score of that substitution across all positions within the same protein, thereby capturing its average functional effect in that protein context. For a given target position, the five substitutions with the most similar functional profiles were selected as nearest neighbors, and the imputed score was computed as the mean of their experimental DMS scores. This variant of the five-nearest-neighbor approach was suggested by the original developers of the 5NN-BLOSUM100 method but has not been previously described in the literature.

To ensure fair comparison among models with differing feature requirements, benchmarking was performed under three complementary evaluation settings. First, all 146 proteins were evaluated using the features available to each respective method. Second, a stringent subset of 40 proteins was analyzed, for which all benchmarked models could be applied to identical datasets, enabling a direct, like-for-like comparison. Third, a subset of 68 proteins was used to enable

additional cross-model comparisons; however, VEFill-max could not be applied to this 68-protein set and was therefore evaluated on the reduced 40-protein subset due to the unavailability of the required EVE scores for some proteins.

To evaluate how imputation performance depends on dataset completeness, we performed a controlled subsampling analysis on 28 high-quality DMS datasets for which all benchmarked models had complete feature availability and overall completeness of at least 94.7%. For each dataset, 10% of variants were held out as a fixed test set, while the remaining variants were subsampled at increasing fractions (10–90% of the full dataset) to simulate partially complete experiments. VEFill dynamically recomputed positional mean DMS scores using only the available training subset at each completeness level, ensuring that no information from held-out variants was used during imputation. Median Pearson correlation across the 28 datasets was used as the summary statistic for all models.

To further assess VEFill's utility in low-data regimes relevant to sparse library design, we evaluated how model performance depends on the number of experimentally measured substitutions available per position. Using the same 28 high-quality DMS datasets employed in the completeness analysis, we trained per-protein VEFill-min and VEFill-max while restricting the training data to a fixed number of mutations per position ($N = 1-20$), randomly sampled for each site. Mean DMS scores were recomputed dynamically using only the available mutations at each position, ensuring that no information from held-out variants was used.

The newly added subsection to the “Results” is provided below:

Benchmarking against existing DMS imputation methods

To assess the practical performance of VEFill relative to other DMS imputation approaches, we compared it against a range of existing methods, including a residue-mean baseline, Wu et al. (2019), Envision, FactorizeDMS, AALasso, and two five-nearest-neighbor approaches, across multiple benchmarking regimes.

To enable fair comparison across models with differing feature requirements, benchmarking was performed across three complementary settings: (i) all 146 proteins with features available per method (Fig. EV3(A)); (ii) a strict 40-protein subset on which all benchmarked models could be applied to exactly the same datasets, enabling direct, like-for-like comparison (Fig. EV3(B)); and (iii) a 68-protein subset that allowed additional cross-model comparison for all methods except VEFill-max, which requires EVE scores that were unavailable for part of

this set (Fig. EV3(C)). Across all evaluation settings, VEFill-min consistently outperformed all other models, while VEFill-max ranked third overall.

To examine how imputation performance depends on dataset completeness, we conducted a controlled subsampling analysis on 28 high-quality DMS datasets with complete feature availability and overall completeness of at least 94.7%, holding out 10% of variants as a fixed test set and subsampling the remaining variants at increasing fractions (10–90%) to simulate partially complete experiments. Across models, distinct trends emerged as data availability increased (Fig. 7). Some approaches, including Wu et al. (2019) and Envision, showed relatively flat performance across completeness levels, whereas FactorizeDMS and AALasso improved gradually as more data became available. In the lowest-data regime (10% training fraction), VEFill-max achieved the highest median Pearson correlation across datasets, while VEFill-min ranked third, slightly below Envision. From 20% completeness onward, both VEFill-min and VEFill-max consistently outperformed all other models. The 5NN-Functional baseline reached comparable performance only after approximately 60% completeness, and from 70% onward its performance became nearly indistinguishable from VEFill.

To further evaluate VEFill's behavior in low-data regimes relevant to sparse library design, we analyzed performance as a function of the number of experimentally measured substitutions per position. Using the same 28 datasets, per-protein VEFill-min and VEFill-max models were trained while restricting the number of mutations per site ($N = 1-20$), with mean DMS scores recomputed dynamically from the available substitutions only. Across datasets, both VEFill-min and VEFill-max showed rapid performance gains as additional substitutions per position became available, with predictive accuracy beginning to plateau at approximately $N \approx 4$ mutations per site (Appendix Fig. S6). Beyond this point, additional measurements yielded diminishing returns. These results indicate that the positional mean DMS score becomes a stable and informative feature once a small number of substitutions are observed at each position, and that VEFill can effectively leverage such sparse positional information in combination with sequence-derived features. Notably, this regime is consistent with the completeness-based analysis (Fig. 7), where VEFill models begin to consistently outperform alternative imputation approaches once approximately 20% of variants are available, corresponding to a similar number of measurements per position. Together, these findings demonstrate that VEFill can support the design and analysis of sparse yet informative variant libraries without requiring full site-saturation mutagenesis.

Additionally, we added text to the “Discussion”:

Across multiple benchmarking analyses, per-protein VEFill demonstrates robust and competitive performance relative to a broad range of existing DMS imputation methods, including residue-wise mean baselines, Wu et al. (2019), Envision, FactorizeDMS, AALasso, and two five-nearest-neighbor approaches. In controlled completeness-based benchmarking, two different VEFill settings consistently outperform all previously published models once at least 20% of variants have been experimentally measured. Importantly, noise-ceiling analyses indicate that per-protein VEFill operates near the accuracy limit imposed by experimental variability in DMS assays, suggesting that further performance gains are primarily constrained by measurement uncertainty rather than model capacity.

2. The authors do not demonstrate the usefulness of VEFill for designing sparse variant libraries in "low-data regimes". However, as discussed in the manuscript and shown by the 0-shot prediction performances, the mean DMS score for a position is the most important feature in the model. Obtaining the mean DMS score requires performing a full DMS experiment, not one using a sparse library. And so to get to the point where one would be able to design a sparse library, one has to build and screen a full library. Is there a minimal number of position-wise score measurements required for the mean DMS score to be informative? Do data from certain substitutions provide more information? This would be informative to determine if the model can help design sparse libraries.

We thank the reviewer for this insightful comment. To address this concern, we have extended the benchmarking analysis in the main manuscript to explicitly evaluate VEFill’s performance in low-data regimes relevant to sparse library design. We added the following text to the “Results” section “Benchmarking VEFill against existing DMS imputation methods” subsection:

To further assess VEFill’s utility in low-data regimes relevant to sparse library design, we evaluated how model performance depends on the number of experimentally measured substitutions available per position. Using the same 28 high-quality DMS datasets employed in the completeness analysis, we trained per-protein VEFill-min and VEFill-max while restricting the training data to a fixed number of mutations per position ($N = 1-20$), randomly sampled for each site. Mean DMS scores were recomputed dynamically using only the available mutations at each position, ensuring that no information from held-out variants was used.

Across datasets, both VEFill-min and VEFill-max showed rapid performance gains as additional substitutions per position became available, with predictive accuracy beginning to plateau at approximately $N \approx 4$ mutations per site (Appendix Fig. S6). Beyond this point, additional measurements yielded diminishing returns. These results indicate that the positional mean DMS score becomes a stable and informative feature once a small number of substitutions are observed at each position, and that VEFill can effectively leverage such sparse positional information in combination with sequence-derived features. Notably, this regime is consistent with the completeness-based analysis (Fig. 7), where VEFill models begin to consistently outperform alternative imputation approaches once approximately 20% of variants are available, corresponding to a similar number of measurements per position. Together, these findings demonstrate that VEFill can support the design and analysis of sparse yet informative variant libraries without requiring full site-saturation mutagenesis.

Additionally, we added text to the “Discussion”:

We find that training per-protein VEFill models using a small, representative set of amino-acid substitutions representing distinct physicochemical classes—specifically histidine, glutamic acid, asparagine, isoleucine, and glycine (H, E, N, I, G)—yields predictive performance comparable to that obtained using randomly selected substitutions at matched training coverage. Together with the observation that per-protein VEFill performance begins to plateau after approximately four mutations per position, these findings suggest that informative yet non-exhaustive variant libraries can be designed by strategically selecting a limited number of substitutions per site. Collectively, this body of evidence supports per-protein VEFill as a practical framework for augmenting partial DMS datasets and guiding efficient experimental design under realistic data constraints.

Regarding which substitutions are most informative, as described in the manuscript, models trained solely on single-nucleotide variants (SNVs) achieved performance comparable to those trained on the full mutation set (Fig. 5D of the main manuscript), indicating that informative SNVs can capture key predictive signals. While exploring whether specific substitution types (e.g., alanine or hydrophobic changes) yield disproportionately informative measurements would be interesting, we feel this lies beyond the scope of the present study due to the combinatorial complexity of exploring the many permutations. However, we do agree that exploring the design of sparse libraries is relevant to this work. Therefore we evaluated a targeted strategy in which the

model was trained using one representative amino acid substitution from each physicochemical class. We added the following text to the “Results” section “Per-protein model evaluation: random splits, LPosO, split by SNV, split by AA class” subsection:

Finally, we evaluated whether training exclusively on a small set of representative amino acid substitutions can provide sufficient predictive signal for accurate imputation. To test this, we selected five substitutions—histidine, glutamic acid, asparagine, isoleucine, and glycine—that each represent a distinct physicochemical class (positive, negative, polar, hydrophobic, and special, respectively). These specific amino acids were chosen because they are closest to the median substitution within their class in the comprehensive analysis of Gray et al. (2017). Models trained exclusively on variants containing one of these five substitutions (H, E, N, I, G) were compared against models trained using a random 23/77 train/test split matched to the average training proportion observed for these subsets. Despite training on a greatly restricted mutation set, models trained only on the (H, E, N, I, G) substitutions achieved performance comparable to the broader 23/77 baseline split (Fig. 5(E)). This result indicates that a carefully selected set of representative substitutions can capture much of the relevant mutational landscape and could serve as a feasible foundation for designing sparse, information-efficient variant libraries.

Additionally, we added text to the “Discussion”:

We find that training per-protein VEFill models using a small, representative set of amino-acid substitutions representing distinct physicochemical classes—specifically histidine, glutamic acid, asparagine, isoleucine, and glycine (H, E, N, I, G)—yields predictive performance comparable to that obtained using randomly selected substitutions at matched training coverage. Together with the observation that per-protein VEFill performance begins to plateau after approximately four mutations per position, these findings suggest that informative yet non-exhaustive variant libraries can be designed by strategically selecting a limited number of substitutions per site. Collectively, this body of evidence supports per-protein VEFill as a practical framework for augmenting partial DMS datasets and guiding efficient experimental design under realistic data constraints.

Other comments:

3. Figure 3: Why are predicted values on average higher? Please scale the x and y axes so they are identical, indicate regression slope, origin, and P-value, and provide a

reference to what a 1 to 1 relationship would look like. This applies to the other scatter plots showing predicted vs measured scores.

We thank the reviewer for noting the upward trend in predictions. After setting identical x/y axis scaling and adding a 1:1 reference line, it becomes clear that the model exhibits the characteristic compression around the mean (Fig. 3). Specifically, the regression slope (0.63) indicates slight overprediction of low-effect variants and underprediction of high-effect variants. The model exhibits mild regression-to-the-mean, whereby extreme observed DMS scores are predicted somewhat closer to the neutral region. This is a common behavior in supervised regression models, especially when the training data distribution is highly enriched for near-neutral variants.

Figure 3. Correlation between predicted and observed DMS scores for the best general cross-protein model.

Predicted scores are from the model using all features except the ESM-1v difference embedding. The 2D histogram shows prediction density. Most predictions cluster around the wild-type score of 1, reflecting the prevalence of near-neutral variants. The normalized DMS scores are theoretically bounded between 0 (nonsense-like) and 1 (neutral-like), but both predicted and observed scores occasionally fall outside this range. This is due to the fact that some experimental DMS datasets contain out-of-bound values, which are preserved during normalization. 1:1 reference line shows perfect prediction ($y = x$).

4. Also Figure 3: Because of the high density of correctly predicted neutral variants, it's possible that the mean RMSE varies substantially along the range of predicted scores. Could the authors provide a binned or rolling RMSE across the full range of predictions?

We agree that local error across the score spectrum is informative. We computed RMSE in quantile-based bins across the full range of both observed and predicted DMS scores (Fig. EV1) and included text to the “Results” section “General cross-protein model evaluation”:

Because the density of variants is highest near the neutral score range, we additionally examined how prediction error varies across the spectrum of DMS scores. We computed RMSE in quantile-based bins across the full range of both observed and predicted normalized DMS scores (Fig. EV1). When binning by observed scores, we observed a characteristic U-shaped error profile: RMSE is lowest in the neutral region, where the training data are densest, and increases toward both the deleterious and high-activity tails. This pattern is consistent with known assay-dependent score compression and increased measurement uncertainty in extreme regions, which reduce the effective dynamic range of DMS experiments.

We further computed RMSE as a function of predicted scores to assess model calibration. In this setting, RMSE is lowest for variants predicted to be neutral, increases for intermediate predicted scores (approximately 0.5–0.7), and decreases again toward the most deleterious predictions. Predicted high-activity variants exhibit error levels comparable to those of mildly deleterious variants. Extreme deleterious and high-activity DMS scores are predicted more conservatively and closer to the neutral region, a behavior that is common in supervised regression models trained on datasets strongly enriched for near-neutral variants. This effect is consistent with the pattern observed in Fig. 3, where predictions exhibit a characteristic compression toward the mean. Together, these results indicate that VEFill is most challenged by intermediate-effect variants, which represent a heterogeneous mixture of functional outcomes and are known to exhibit higher assay noise. In contrast, VEFill provides reliable quantitative estimates for mildly deleterious and mildly high-activity variants, resulting in RMSE values comparable to those observed in the neutral regime. Predictions at the extreme tails should therefore be interpreted as conservative approximations rather than precise estimates of extremal effects.

Figure EV1. RMSE computed in quantile-based bins using either observed (blue) or predicted (red) DMS scores. The observed-binned curve reveals variation in predictive accuracy across the biological effect spectrum, while the predicted-binned curve reflects model calibration across its output range.

5. Also Figure 3: Then, based on these RMSE distributions, are the authors confident that VEFill can provide reliable score imputation beyond determining if a variant is benign?

We thank the reviewer for this comment. The binned RMSE analyses (Fig. X4) show that VEFill achieves its lowest error in the neutral region and maintains comparable RMSE in both the mildly deleterious and mildly hyperactive tails. This reflects the model's conservative regression-to-the-mean behavior, whereby extreme effects are predicted only when strongly supported by the features, avoiding unreliable extrapolation. The only region with clearly elevated error is the intermediate-effect range, which is known to exhibit the greatest assay noise and biological heterogeneity in DMS datasets. Because VEFill retains stable error across the potentially deleterious and hyperactive ranges and avoids large tail deviations, we are confident that it provides meaningful quantitative imputations beyond simple benign/neutral classification.

6. (Finally) Also Figure 3: DMS assays can be quite noisy. Does the error in predictions scale with DMS score measurement error? If so, would the model benefit from excluding low confidence scores? Alternatively, could the model also be used to flag positions where a DMS score with high experimental error has a high chance of being erroneous?

We thank the reviewer for these important questions. A detailed analysis of measurement noise, its impact on achievable predictive accuracy, and the effect of excluding high-uncertainty variants is now provided in our response to Reviewer #1

(comment #3). Briefly, we show that VEFill operates close to the noise-imposed accuracy ceiling and that excluding up to 30% of the highest-uncertainty variants does not materially improve per-protein performance. While removing noisy points does not enhance accuracy, large discrepancies between VEFill predictions and experimentally reported scores—particularly for variants with high assay uncertainty—may still help flag measurements that merit further validation. Please see our full response to Reviewer #1 for further detail.

7. Figure 4b: Are the best predicted domains shown here from overrepresented domain families in the human domainome dataset (as mentioned later in the text)?

We thank the reviewer for this question. In Fig. 4(B) we evaluated per-protein performance using a leave-one-protein-out (LOPO) model trained on the 140-domain dataset, which contains 57 distinct Pfam families. We examined whether the top-performing proteins in this analysis belong to Pfam families that are more heavily represented in the training data.

Among the 10 best-predicted domains, the corresponding Pfam families are represented in the 140-domain dataset by 4, 1, 8, 11, 5, 5, 1, 1, 4, and 2 domains, respectively (ordered from highest to lowest R^2). Thus, while some of the top-performing proteins do come from moderately represented families, others belong to Pfam families represented by only a single domain. Overall, there is no clear trend indicating that the best-predicted domains are drawn disproportionately from overrepresented Pfam families.

8. Table 1: The model shows high variability in terms of performance, even within assay categories. With the high number of datasets available in MaveDB, is there any reason a more systematic evaluation was not performed? This might allow for a better understanding of scenarios where the model is expected to perform well or not. This is especially true of the DMSscore+ESMv1 embeddings model, which requires very little data on the protein sequence of interest.

We thank the reviewer for this thoughtful comment. Our evaluation (Table 1) was designed to test whether a model trained on Human Domainome 1 generalizes to biophysically similar external assays, rather than to benchmark the model across all of MaveDB. We therefore focused on a small set of well-characterized representative assays spanning both stability and activity. A systematic evaluation across a curated subset of compatible MaveDB datasets is indeed valuable, but requires careful assay harmonization and is beyond the scope of the present work. We view this as an important direction for future work. We added the following text to the “Discussion”:

Evaluation on both internal and external datasets confirmed VEFill's capacity to generalize across previously unseen domains assayed under stability-focused experimental conditions. However, its performance on activity-based datasets was markedly reduced, likely due to mismatches between training and test assay modalities, as well as the complex nature of cellular phenotypes. A systematic evaluation across a curated subset of compatible MaveDB datasets would be valuable for further characterizing model performance across assay types, but would require careful assay harmonization.

Notably, for the per-protein version of VEFill, we did evaluate the model on a larger collection of MaveDB datasets (77 external datasets) as a part of our benchmarking; however, that differs conceptually because a per-protein model is trained for each protein rather than applying a single pretrained model.

9. Also Table 1: For some proteins in MAVEdb (like PTEN), both variant abundance and activity data are available for the same DMS library. Have the authors compared model performance on both?

We thank the reviewer for this helpful comment. For PTEN, we indeed used two assays - one measuring variant abundance (VAMP-seq) and one measuring lipid phosphatase activity. Although both target the same protein, they were generated by different studies, using different experimental platforms, different scoring schemes, and different normalization procedures. They do not represent two readouts from the same DMS library, and therefore cannot be directly compared as paired measurements from a shared experiment.

Our goal was to evaluate whether a model trained on Human Domainome 1 can generalize across independent external assays, not to compare performance across multiple readouts from the same library. For PTEN, we therefore evaluated VEFill separately on the abundance dataset (Matreyek et al. 2021) and on the activity dataset (Mighell et al. 2018), each treated as an independent external benchmark. This allowed us to assess model applicability across fundamentally different assay types. Consistent with our broader findings, VEFill generalized well to stability-based abundance measurements but showed reduced performance on the activity assay

A paired analysis - where abundance and activity measurements are generated from the same mutagenesis library under matched conditions - would indeed be valuable, but such datasets are rare and would benefit from a bespoke modelling approach that is aware of the two different assay readouts. We think this would be an interesting future direction for modelling DMS data, but is out of scope for this study.

10. To improve manuscript clarity, I would suggest reducing the number of main text figures and providing data visualization with more depth. For example, Figure 6 presents 4 different heatmaps, which could be condensed into two scatter plots, with labels showing discrepancies between LOPosO and LOVarO.

We thank the reviewer for this helpful suggestion. In the revised manuscript, we replaced the four heatmaps originally shown in Fig. 6 with the pair of LOPosO-LOVarO scatter plots, which provide a clearer and more compact comparison between the two evaluation strategies. These plots include labels for substitutions with the largest discrepancies, making the differences between LOPosO and LOVarO easier to interpret. The full heatmaps are now provided in Fig. EV2, where they preserve positional detail without burdening the main text. We elected to retain the other multi-panel figures (for example, Fig. 5) because each panel conveys a distinct, non-redundant aspect of model behavior - such as performance across data-splitting strategies, feature subsets, or amino-acid class restrictions - that would be lost if collapsed into a single visualization.

Figure 6. Comparison of LOPosO and LOVarO fine-grained prediction errors for two representative proteins.

(A) FADD death effector domain and **(B)** TRIM44 B-box-type zinc-finger domain. Each scatter plot compares variant-level prediction errors obtained under the leave-one-position-out (LOPosO) and leave-one-variant-out (LOVarO) strategies. Points represent individual single-amino-acid substitutions; the diagonal line indicates identical error between the two strategies. Labels highlight the top-10 substitutions with the largest discrepancies between LOPosO and LOVarO squared error, indicating cases where position-level versus variant-level generalization differs most strongly.

17th Feb 2026

Manuscript Number: MSB-2025-13319R

Title: VEFill: accurate and generalizable deep mutational scanning score imputation across protein domains

Author: Polina Polunina

Wolfgang Maier

Alan Rubin

Dear Dr Rubin,

Thank you for the submission of your revised manuscript to Molecular Systems Biology. We have now received the enclosed reports from the referees that were asked to re-assess it. As you will see the reviewers are now globally supportive and I am pleased to inform you that we will be able to accept your manuscript pending the following final amendments:

- 1) In the main manuscript file, please format the Data availability section according to the example below.
"The datasets and computer code produced in this study are available in the following databases:
- Chip-Seq data: Gene Expression Omnibus GSE46748 (<https://www.ncbi.nlm.nih.gov/geo/query/acc.cgi?acc=GSE46748>)
- Modeling computer scripts: GitHub (<https://github.com/SysBioChalmers/GECKO/releases/tag/v1.0>)
- [data type]: [full name of the resource] [accession number/identifier] ([doi or URL or identifiers.org/DATABASE:ACCESSION])"
- 2) Please ensure that the Zenodo link for Polunina 2025b is correct/publicly available, as we were not able to find the associated datasets.
- 3) Please rename "Competing Interest" to "Disclosure and competing interests statement". We updated our journal's competing interests policy in January 2022 and request authors to consider both actual and perceived competing interests. Please review the policy <https://link.springer.com/partners/embo-press/editorial-policies#Competing%20interest%20disclosures> and update your competing interests if necessary.
- 4) References: DOIs should only be used for preprints and datasets that have not been published yet. Please update the reference list accordingly.
- 5) Please place individual sections of the manuscript in the following order: Title page - Abstract & Keywords - Introduction - Results - Discussion - Methods - Data Availability - Acknowledgements - Disclosure and Competing Interests Statement - References - Figure Legends - (Main Tables with legends if applicable) - Expanded View Figure Legends.
- 6) Please ensure that the missing 'Figure Legends' and 'Expanded View Figure Legends' section titles are included in the main manuscript.
- 7) For the figures and figure legends, please take care of the following:
 - Please make sure to update the callouts of all figures in the main manuscript text. Currently figure callouts are missing for Figure missing for Figure 6A-B and Figure 8A-B.
 - Please note that the exact p values are not provided in the legend of figure 3
 - Please indicate the statistical test used for data analysis in the legend of figure 3
 - Please note that the box plots need to be defined in terms of minima, maxima, centre, bounds of box and whiskers, and percentile in the legend of figure 4A
- 8) Tables: Please upload each EV table as one .xsl file per table and rename them to Dataset EV1-4. Each dataset will need its legend removed from the manuscript and added to the corresponding file in a separate tab. Please update their callouts in main manuscript text.
- 9) Appendix file: Please remove the author list and affiliations from the title page of the Appendix.
- 10) Funding: Please note that funding information should be given in the "Acknowledgements" section (not in its own separate section). Please ensure that all funding sources are entered into the manuscript submission system without using the comments box.
- 11) Source Data: Please ensure that a completed Source Data checklist is uploaded as a Related Manuscript File. The checklist and instructions should have been sent to you after the previous decision letter to revise your manuscript. If you have any Source Data, it should be organized as a single source data file (zipped) per figure for main figures (all EV and/or Appendix figure Source Data can be included in a single folder), with the panels clearly visible in the folder structure instead of a single excel file for all Source Data. e.g. all the Source data files for figure 1 need to be saved in a single folder and this needs to be zipped and then uploaded as "SD figure 1.zip" file.
- 12) As part of the EMBO Publications transparent editorial process initiative (see our policy here: https://www.embopress.org/transparent-process#Review_Process), Molecular Systems Biology will publish online a Peer Review File (PRF) to accompany accepted manuscripts. This file will be published in conjunction with your paper and will include the anonymous referee reports, your point-by-point response and all pertinent correspondence relating to the manuscript. Let us know whether you agree with the publication of the PRF and as here, if you want to remove or not any figures from it prior to publication. Please note that the Authors checklist will be published at the end of the PRF.
- 13) After your paper is published, we may promote it on social media. If you have any handles or hashtags for Bluesky you would like included, please let us know.
- 14) Please provide a point-by-point letter INCLUDING my comments and your detailed responses (as Word file).

I look forward to reading a new revised version of your manuscript as soon as possible.

Yours sincerely,

Poonam Bheda, PhD
Scientific Editor
Molecular Systems Biology

Reviewer #1:

I thank the authors for a thorough revision, and I have no additional concerns.

Reviewer #2:

The authors have adequately addressed all my comments.

17th Feb 2026

Manuscript Number: MSB-2025-13319R

Title: VEFill: accurate and generalizable deep mutational scanning score imputation across protein domains

Author: Polina Polunina

Wolfgang Maier

Alan Rubin

Dear Dr Rubin,

Thank you for the submission of your revised manuscript to Molecular Systems Biology. We have now received the enclosed reports from the referees that were asked to re-assess it. As you will see the reviewers are now globally supportive and I am pleased to inform you that we will be able to accept your manuscript pending the following final amendments:

1) In the main manuscript file, please format the Data availability section according to the example below.

"The datasets and computer code produced in this study are available in the following databases:

- Chip-Seq data: Gene Expression Omnibus GSE46748

(<https://www.ncbi.nlm.nih.gov/geo/query/acc.cgi?acc=GSE46748>)

- Modeling computer scripts: GitHub

(<https://github.com/SysBioChalmers/GECKO/releases/tag/v1.0>)

- [data type]: [full name of the resource] [accession number/identifier] ([doi or URL or identifiers.org/DATABASE:ACCESSION)]"

We have reformatted the Data Availability section to follow the requested structured format. The section now begins with the prescribed introductory sentence and lists each dataset and code repository in bullet-point format, including names of the resource, repository names, version identifiers, citation and DOI/URLs where appropriate.

2) Please ensure that the Zenodo link for Polunina 2025b is correct/publicly available, as we were not able to find the associated datasets.

We have verified that the Zenodo record corresponding to Polunina 2025b (VEFill v1.0.1 frozen release) is publicly accessible and that the DOI resolves correctly. The

Data Availability section has been updated accordingly to ensure clarity and reproducibility.

3) Please rename "Competing Interest" to "Disclosure and competing interests statement". We updated our journal's competing interests policy in January 2022 and request authors to consider both actual and perceived competing interests. Please review the policy <https://link.springer.com/partners/embo-press/editorial-policies#Competing%20interest%20disclosures> and update your competing interests if necessary.

The section has been renamed to "Disclosure and competing interests statement" as requested. We have reviewed the updated EMBO Press policy and confirm that the statement reflects both actual and perceived competing interests. No additional competing interests apply.

4) References: DOIs should only be used for preprints and datasets that have not been published yet. Please update the reference list accordingly.

We have checked that all DOIs in the references refer to preprints or datasets. The previous DOI citation for "MaveDB" referred to the versioned bulk data download. This Zenodo record has now been updated and the new name is used here for clarity. We have also replaced the Zenodo DOI reference for the pandas software package with the conference proceedings reference recommended by the package authors.

5) Please place individual sections of the manuscript in the following order: Title page - Abstract & Keywords - Introduction - Results - Discussion - Methods - Data Availability - Acknowledgements - Disclosure and Competing Interests Statement - References - Figure Legends - (Main Tables with legends if applicable) - Expanded View Figure Legends.

The manuscript has been reorganized to match the required order.

6) Please ensure that the missing 'Figure Legends' and 'Expanded View Figure Legends' section titles are included in the main manuscript.

Both section titles have now been explicitly included in the main manuscript.

7) For the figures and figure legends, please take care of the following:

- Please make sure to update the callouts of all figures in the main manuscript text. Currently figure callouts are missing for Figure missing for Figure 6A-B and Figure 8A-B.

We have updated the manuscript text to include callouts for Figures 6(A–B) and 8(A–B) at the appropriate locations.

- Please note that the exact p values are not provided in the legend of figure 3

The legend for Figure 3 has been updated to explicitly report the correlation coefficient and associated p-value.

- Please indicate the statistical test used for data analysis in the legend of figure 3

The legend now specifies that statistical significance was assessed using a two-sided Pearson correlation test (via ordinary least squares linear regression).

- Please note that the box plots need to be defined in terms of minima, maxima, centre, bounds of box and whiskers, and percentile in the legend of figure 4A

The Figure 4(A) legend now defines all box plot components explicitly.

8) Tables: Please upload each EV table as one .xsl file per table and rename them to Dataset EV1-4. Each dataset will need its legend removed from the manuscript and added to the corresponding file in a separate tab. Please update their callouts in main manuscript text.

Each EV table has been uploaded as a separate .xls file (Dataset EV1–EV4). Legends have been added to a separate tab within each corresponding dataset file. Callouts in the main text have been updated accordingly.

9) Appendix file: Please remove the author list and affiliations from the title page of the Appendix.

The author list and affiliations have been removed from the Appendix title page.

10) Funding: Please note that funding information should be given in the "Acknowledgements" section (not in its own separate section). Please ensure that all funding sources are entered into the manuscript submission system without using the comments box.

The separate Funding section has been removed. All funding information has been merged into the Acknowledgements section. Funding sources have also been entered into the manuscript submission system as requested.

11) Source Data: Please ensure that a completed Source Data checklist is uploaded as a Related Manuscript File. The checklist and instructions should have been sent to you after the previous decision letter to revise your manuscript. If you have any Source Data, it should be organized as a single source data file (zipped) per figure for main figures (all EV and/or Appendix figure Source Data can be included in a single folder), with the panels clearly visible in the folder structure instead of a single excel file for all Source Data. e.g. all the Source data files for figure 1 need to be saved in a single folder and this needs to be zipped and then uploaded as "SD figure 1.zip" file.

We have uploaded the completed Source Data checklist as a Related Manuscript File. Source Data files have been organized as requested, with one zipped folder per figure (e.g., "SD Figure 2.zip"), containing clearly labeled panel-specific files.

We opted to provide individual zipped folders for each EV and Appendix figure, since some Source Data files are associated with a main figure and also an EV figure.

12) As part of the EMBO Publications transparent editorial process initiative (see our policy here: https://www.embopress.org/transparent-process#Review_Process), Molecular Systems Biology will publish online a Peer Review File (PRF) to accompany accepted manuscripts. This file will be published in conjunction with your paper and will include the anonymous referee reports, your point-by-point response and all pertinent correspondence relating to the manuscript. Let us know whether you agree with the publication of the PRF and as here, if you want to remove or not any figures from it prior to publication. Please note that the Authors checklist will be published at the end of the PRF.

We agree with the publication of the Peer Review File and do not request removal of any figures.

13) After your paper is published, we may promote it on social media. If you have any handles or hashtags for Bluesky you would like included, please let us know.

We have no specific social media handles to include.

14) Please provide a point-by-point letter INCLUDING my comments and your detailed responses (as Word file).

This document constitutes our detailed point-by-point response as requested.

I look forward to reading a new revised version of your manuscript as soon as possible.

Thank you. Please note that all revision highlights (previously marked in yellow using \hl{ }) have been removed from the manuscript to provide a clean final version.

Yours sincerely,

Poonam Bheda, PhD
Scientific Editor
Molecular Systems Biology

Reviewer #1:

I thank the authors for a thorough revision, and I have no additional concerns.

We thank the reviewer for the careful evaluation and supportive assessment.

Reviewer #2:

The authors have adequately addressed all my comments.

We thank the reviewer for their constructive feedback and positive evaluation.

6th Mar 2026

Manuscript number: MSB-2025-13319RR

Title: VEFill: accurate and generalizable deep mutational scanning score imputation across protein domains

Dear Dr Rubin,

Thank you again for sending us your revised manuscript with the requested modifications. I am pleased to inform you that your paper has been accepted for publication at Molecular Systems Biology.

You may qualify for financial assistance for your publication charges - either via a Springer Nature fully open access agreement or an EMBO initiative. Check your eligibility: <https://link.springer.com/journal/44320/how-to-publish-with-us>

Yours sincerely,

Poonam Bheda, PhD
Scientific Editor
Molecular Systems Biology

>>> Please note that it is Molecular Systems Biology policy for the transcript of the editorial process (containing referee reports and your response letter) to be published as an online supplement to each paper. If you do NOT want this, you will need to inform the Editorial Office via email immediately. More information is available here: <https://link.springer.com/partners/embo-press/editorial-policies#Peer%20review>